# Energy-based Potential Games
# for Joint Motion Forecasting and Control

**Christopher Diehl**[1]    **Tobias Klosek**[1]    **Martin Krüger**[1]
**Nils Murzyn**[2]    **Timo Osterburg**[1]    **Torsten Bertram**[1]
[1]TU Dortmund University    [2]ZF Friedrichshafen AG, Artificial Intelligence Lab
`forename.surename@tu-dortmund.de`    `nils.murzyn@zf.com`
`https://github.com/rst-tu-dortmund/diff_epo_planner`

**Abstract:** This work uses game theory as a mathematical framework to address interaction modeling in multi-agent motion forecasting and control. Despite its interpretability, applying game theory to real-world robotics, like automated driving, faces challenges such as unknown game parameters. To tackle these, we establish a connection between differential games, optimal control, and energy-based models, demonstrating how existing approaches can be unified under our proposed *Energy-based Potential Game* formulation. Building upon this, we introduce a new end-to-end learning application that combines neural networks for game-parameter inference with a differentiable game-theoretic optimization layer, acting as an inductive bias. The analysis provides empirical evidence that the game-theoretic layer adds interpretability and improves the predictive performance of various neural network backbones using two simulations and two real-world driving datasets.

## 1 Introduction

Modeling multi-agent interactions is essential for many self-driving vehicle (SDV) applications like motion forecasting and control. For instance, an SDV has to interact with other pedestrians or human-driven vehicles to navigate safely toward its goal locations, while interaction outcomes can be inherently multi-modal. As an illustration, consider a highway merge where one agent may the other agent pass or merge first (Fig. 1 (a)). This work addresses modeling these different future evolutions, which we call *modes*. Hence, we aim to estimate a conditional distribution $p(\mathbf{X}|\mathbf{o})$, defined by random variables of future joint trajectories $\mathbf{X}$ and observations $\mathbf{o}$ (e.g., agent histories and a map).

There are two dominant groups of methods to model these interactions. Game-theoretic approaches [1] incorporate priors based on physics and rationality, such as system dynamics and agent preferences, into interaction modeling, and ensure the interpretability of latent variables in the models [2]. Here, non-cooperative game-theoretic equilibria describe interactions, and solvers (*implicit* functions) typically search for local equilibria [3, 4], resulting in a single (uni-modal) joint strategy $\mathbf{u}$. Given system dynamics (e.g., vehicle models), we can compute the resulting future joint state trajectory $\mathbf{x}$ based on $\mathbf{u}$. While finding cost parameters for an SDV is non-trivial [5, 6], knowing the preferences and goals of all other agents is an unrealistic assumption. For example, the intents of human drivers are not directly observable. That makes online inference of game parameters necessary [7].

On the other hand, works like [10] and [11] employ *explicit strategies* by utilizing feed-forward neural networks (*explicit* functions) with parameters $\theta$ to generate multi-modal joint strategies $\mathbf{U} = F_\theta(\mathbf{o})$ based on observation $\mathbf{o}$. Although neural network-based approaches represent the state-of-the-art (SOTA) on motion forecasting benchmarks, they are considered low interpretable black-box models with limited controllability [12]. *How can we leverage the benefits of both groups of approaches?*

Energy-based neural networks [13] provide an *implicit* mapping $\mathbf{u}^* = \arg\min_{\mathbf{u}} E_\theta(\mathbf{u}, \mathbf{o})$ and [14] demonstrates the advantageous properties of such implicit models in single-agent control experiments. This work shows how to parameterize the energy $E_\theta(\cdot)$ with a potential game formulation [15].

7th Conference on Robot Learning (CoRL 2023), Atlanta, USA.

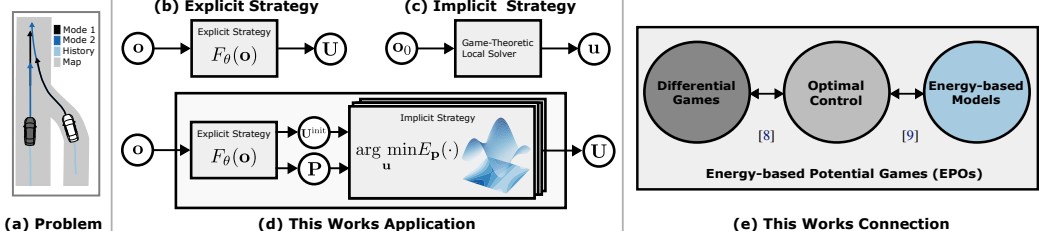

Figure 1: Problem, strategy representations, and the proposed connection. (a) We aim to predict multi-modal future trajectories $\mathbf{X}$ (black, dark blue), derived from joint strategies $\mathbf{U}$ based on observations $\mathbf{o}$. (b) Learned explicit strategy producing a multi-modal solution $\mathbf{U}$. (c) Implicit game-theoretic strategy, only considering the current observation $\mathbf{o}_0$, with fixed game parameters, yielding a local (uni-modal) solution $\mathbf{u}$. (d) We infer multi-modal strategy initializations $\mathbf{U}^{\text{init}}$ and game parameters $\mathbf{P} = \{\mathbf{p}^1, \ldots, \mathbf{p}^M\}$ with an explicit strategy, then perform $M$ game-theoretic energy minimizations in *parallel* in a learnable end-to-end framework. (e) EPOs as a connection of different research areas.

Hence, we combine *explicit strategies* for initialization and game parameter inference with *implicit strategies*, as shown in Fig. 1 (d).

**Contribution.**[1] This work proposes *Energy-based Potential Games* (EPOs) as a class of methods connecting differential games, optimal control, and energy-based models (EBMs) (Fig. 1 (e)). We show how to derive an EBM formulation starting from a differential game. Further, this work proposes a new differentiable *Energy-based Potential Game Layer* (EPOL), which is combined with hierarchical neural networks in a novel system architecture for multi-agent forecasting and control. Third, we demonstrate that this approach improves different neural networks in simulated and real-world motion forecasting experiments and is applicable for motion control of a single-agent.

## 2 Related Work

**Game-Theoretic Planning.** Game-theoretic motion planning methods [17, 3, 4], aiming to find Nash equilibria (NE), capture the interdependence about how one agent's action influences other agents' futures. However, these approaches typically involve computationally intensive coupled optimal control problems. Hence, [2] and [18] formulate the problem as a potential game [15], reducing the formulation to only a single optimal control problem (OCP). Filtering techniques [19] and inverse game solvers [7] are used to learn parameters. However, these methods have been primarily evaluated in simulation, and the learning objectives of [7] assumes uni-modal distributions. In contrast, our work utilizes neural networks to infer multi-modal parameters and initializations and learns end-to-end.

**Data-driven Motion Forecasting.** Motion forecasting with neural networks currently represents the SOTA in benchmarks for human vehicle [20] or pedestrian prediction [21]. Most works focus on modeling interactions in the observation encoding part. For instance, the attention [11, 22, 23] or convolutional social pooling [24] mechanism is commonly employed. Some works [25, 26] use latent variable models to generate multi-modal joint predictions. In general, our approach is complementary to these developments, as it can be integrated on top of various network architectures (Sec. 5.1).

**Energy-based Models.** Reviews for EBMs are provided in [13, 27], and [28] identifies three main paradigms for energy learning: (i) Conditional Density Estimation (CDE), (ii) Exact Energy Minimization, (iii) Unrolled Optimization (UN). Models of the first group (i) use the probabilistic interpretation that low-energy regions have high probability. Approaches utilize maximum likelihood estimation (MLE) [27], noise contrastive (NC) divergence [29], or NC estimation [30] objectives. Type (ii) methods solve the energy optimization problem exactly and differentiate using techniques such as the *implicit function theorem* (IFT) [31]. Methods from type (iii), like [28], approximate the solution with a finite number of gradient steps and backpropagate through the unrolled optimization.

**Differentiable Optimization.** Advances in differentiable optimization [31, 32] enable our work, to combine optimization problems with learning-based models such as neural networks. [33, 34] introduce differentiable architectures using non-game-theoretic motion planners for SDVs. The

---

[1]A prior version of this work was presented at a non-archival workshop [16].

concurrent work of [4] proposes a combination of a differentiable joint planner with a simple network architecture (multilayer perceptron (MLP)) and serves as a proof of concept. Further, [4] evaluates open-loop on a simulated dataset and does not account for multi-modality. In contrast, we produce multi-modal predictions, evaluate with real-world datasets, and use different SOTA neural networks.

**Discussion of Related Applications.** Tab. 1 shows how existing applications from the field of multi-agent forecasting can be viewed under the new EPO framework (Sec. 3) by comparing energy structures, the method for solving the energy optimization problem (6), and learning types. These works provide additional empirical evidence that modeling real-world multi-agent interactions as EPOs is promising. DS-DNet [36] and JFP [35] use a neural network

Table 1: A comparison of EPO applications.

|  | Energy Structure | Energy Optimization | Learn. Type |
|---|---|---|---|
| JFP [35]* | NN | SB (LB) | CDE |
| DSDNet [36] | Nonlin.+NN | SB (MB) | CDE |
| TGL [2]⋄ | Convex | GB | IFT |
| EBIOC [9]⋆ | Nonlin./NN | GB | CDE |
| Sec. 4 (Ours) | Nonlin. | GB | UN |

*Assuming control prediction of the backbone [11] and the fully-connected graph of [35] ⋄ [2] uses hard inequality constraints. EPOs penalize inequalities in the energy. ⋆ Assuming the costs in [9] define a potential game [18, Theorem 2].

(NN) to approximate the energy and perform sampling-based (SB) optimization. The sampled future states can be generated by unrolling the dynamics (Eq. 1) with controls obtained from model-based (MB) or learning-based (LB) sampling. Both works use the probabilistic interpretation of EBMs with a conditional density given by $p_\theta(\mathbf{u}|\mathbf{o}) = \frac{1}{Z}\exp(-E_\theta(\mathbf{u},\mathbf{o}))$ and normalization constant $Z$, which is often intractable to compute in closed form. TGL [2] uses the IFT to learn the energy, which requires convergence to an optimal solution [32]. The energy is a linear combination of energy features $c_j(\cdot)$, given by $E_{\mathbf{w}}(\mathbf{u},\mathbf{o}) = \sum_j \mathbf{w}_j c_j(\mathbf{o},\mathbf{u})$, with $w_j$ as an inferred weight. The approach uses gradient-based convex minimization, whereas the required convexity of $c$ is restrictive for general real-world scenarios (e.g., curvy lanes). Neural networks as energy approximations are more expressive and can overcome the design of features. However, linear combinations of features allow the incorporation of domain knowledge into the training process and provide a level of interpretability [12, 4]. EBIOC [9] proposes to use EBMs for inverse optimal control with type (i) learning. Unlike [9], we use SOTA neural network structures with a type (iii) formulation and provide deeper analysis in multi-agent scenarios and multi-modal solutions. To the author's best knowledge, besides the new connection, our application is the first to combine nonlinear differentiable game-theoretic optimization with neural networks and successfully demonstrate its performance on considerably large real-world datasets.

## 3 Energy-based Potential Games

### 3.1 Background

**Differential Games.** Assume we have $N$ agents, and $\mathbf{u}_i(t) \in \mathbb{R}^{n_{u,i}}$ represents the *control* vector, and $\mathbf{x}_i(t) \in \mathbb{R}^{n_{x,i}}$ the *state* vector for each agent $i, 1 \le i \le N$ at timestep $t$ with dimensions $n_{u,i}$ and $n_{x,i}$. The joint state evolves according to a time-continuous differential equation:

$$\dot{\mathbf{x}}(t) = f(\mathbf{x}(t), \mathbf{u}(t), t), \tag{1}$$

with dynamics $f(\cdot)$ and starting at the initial state $\mathbf{x}(0) = \mathbf{x}_0$. Let $\mathbf{u}(t) = (\mathbf{u}_1(t), \cdots, \mathbf{u}_N(t)) \in \mathbb{R}^{n_u}$ and $\mathbf{x}(t) = (\mathbf{x}_1(t), \cdots, \mathbf{x}_N(t)) \in \mathbb{R}^{n_x}$ be the concatenated vectors of all agents controls and states at time $t$, with dimensions $n_u$ and $n_x$. Assume each agent minimizes cost

$$C_i(\mathbf{x}_0, \mathbf{u}) = \int_0^T L_i(\mathbf{x}(t), \mathbf{u}(t), t)\mathrm{d}t + S_i(\mathbf{x}(T)) \tag{2}$$

with time horizon $T$, running cost $L_i$ and terminal cost $S_i$ of agent $i$. Let $\mathbf{u}_i : [0,T] \times \mathbb{R}^{n_{x,i}} \to \mathbb{R}^{n_{u,i}}$ define an *open-loop strategy*[2] and $\mathbf{u}_{-i}$ defines the open-loop strategy for all players *except* $i$. Then, $\mathbf{u}$ defines a *joint strategy* for all agents. We can now characterize the differential game with notation: $\Gamma_{\mathbf{x}_0}^T := (T, \{\mathbf{u}_i\}_{i=1}^N, \{C_i\}_{i=1}^N, f)$. Let $(\mathbf{u}_i, \mathbf{u}_{-i}^*)$ be a shorthand for $(\mathbf{u}_1^*, \ldots, \mathbf{u}_{i-1}^*, \mathbf{u}_i, \mathbf{u}_{i+1}^*, \ldots, \mathbf{u}_N^*)$. Intuitively speaking, in a NE, no agent is incentivized to unilaterally change its strategy, assuming that other agents keep their strategy unchanged. We can formalize this by recalling the definition of NE from [1, Chapter 6]:

---

[2]Open-loop strategies provide equivalence between strategy and control actions for all time instants [1]. Hence, for clarity, we omitted to introduce a new variable for the strategy, and overloaded the notation for $\mathbf{u}_i$ such that it describes the controls of agent $i$ in the time interval $[0,T]$.

**Definition 3.1.** Given a differential game defined by all agents dynamics (1), and costs (2), a joint strategy $\mathbf{u}^* = (\mathbf{u}_1^*, \ldots, \mathbf{u}_n^*)$ is called an open-loop Nash equilibrium (OLNE) if, for every $i = 1, \ldots, N$ the following conditions hold: $C_i (\mathbf{x}_0, \mathbf{u}^*) \leq C_i (\mathbf{x}_0, \mathbf{u}_i, \mathbf{u}_{-i}^*) \; \forall \mathbf{u}_i$.

**Potential Differential Games.** Finding a NE involves solving *N-coupled* OCPs, which is non-trivial and computationally demanding [2, 18]. However, according to [8], there exists a class of games, namely *potential differential games* (PDGs), in which only the solution of a *single* OCP is required, and its solutions correspond to OLNE of the original game.

**Definition 3.2.** (cf. [8]) A differential game $\Gamma_{x_0}^T$, is called an open-loop PDG if there exists an OCP such that an open-loop optimal solution of this OCP is an OLNE for $\Gamma_{x_0}^T$.
[18, Theorem 1] (App. C) implies that such an OCP is given by:

$$\min_{\mathbf{u}(\cdot)} \int_0^T p(\mathbf{x}(t), \mathbf{u}(t), t)\mathrm{d}t + \bar{s}(\mathbf{x}(T)) \quad \text{subject to } \dot{\mathbf{x}}_i(t) = f(\mathbf{x}_i(t), \mathbf{u}_i(t), t) \quad \forall i, \qquad (3)$$

under the assumption of decoupled dynamics. Here, $p(\cdot)$ and $\bar{s}(\cdot)$ are so called *potential functions*. It is further shown that the potential function cost terms of the agents have to be composed of two terms: (i) Cost terms $C_i^{\mathrm{own}}(\mathbf{x}_i(t), \mathbf{u}_i(t))$ that only depend on the state and control of agent $i$ and (ii) pair-wise coupling terms $C_{i,j}^{\mathrm{pair}}(\mathbf{x}_i(t), \mathbf{x}_j(t))$ between agents $i$ and $j$. Further, the coupling terms have to fulfill the property [18, Theorem 2]: $C_{i,j}^{\mathrm{pair}}(\mathbf{x}_i(t), \mathbf{x}_j(t)) = C_{j,i}^{\mathrm{pair}}(\mathbf{x}_i(t), \mathbf{x}_j(t)) \forall i \neq j$. Intuitively speaking, two agents $i$ and $j$ care the same for common social norms. Then $p(\cdot)$ and $s(\cdot)$ are given by

$$p(\cdot) = \sum_{i=1}^N C_i^{\mathrm{own}}(\mathbf{x}_i(t), \mathbf{u}_i(t)) + \sum_{1 \leq i < j}^N C_{i,j}^{\mathrm{pair}}(\mathbf{x}_i(t), \mathbf{x}_j(t)) \quad \text{and} \quad \bar{s}(\cdot) = \sum_{i=1}^N C_{i,T}^{\mathrm{own}}(\mathbf{x}_i(T)). \quad (4)$$

*SDV Running Example: The cost function $C_i$ may incorporate agents' goal-reaching objectives ($S_i$), while the running costs $L_i$ account for collision avoidance and aim to minimize control efforts. Further, possible cost functions are given in [37]. $\dot{\mathbf{x}}_i(t) = f(\mathbf{x}_i(t), \mathbf{u}_i(t), t)$ represents the traffic participant dynamics (e.g., vehicles or pedestrians), characterized by bicycle, unicycle [38], or integrator dynamics [10]. The cost terms $C_i^{own}$ then describe the control input cost and goal-reaching cost, whereas $C_{i,j}^{pair}$ describes common social norms, such as collision avoidance. Assuming social norms remain reasonable since drivers typically intend to avoid collisions in the majority of scenarios [18, 2] and assertive drivers can be modeled with high weights of other terms (e.g., goal-reaching).*

### 3.2 Connecting Potential Differential Games and Energy-based Models

While PDGs provide more tractable solutions to the game, challenges still arise due to unknown game parameters. Hence, we aim to infer the parameters online using function approximators based on an observed context $\mathbf{o}$. As a solution, we now demonstrate how to connect PDGs to EBMs.

**Direct Transcription.** Due to its simplicity and resulting low number of optimization variables, we apply single-shooting, a direct transcription method [39], to transform the time-continuous formulation of (3) into a discrete-time OCP. Let the discretized time interval be $[0, T]$ with $0 = t_0 \leq t_1 \leq \cdots \leq t_k \leq \cdots \leq t_K = T$ and $k = 0, \ldots, K$. We assume a piecewise constant control $\mathbf{u}_i(t_k) := \mathbf{u}_{i,k} = \text{constant}$ for $t \in [t_k, t_k + \Delta t)$, where $\Delta t = t_{k+1} - t_k$ denotes the time interval. Then $\mathbf{x}_{i,k+1} = f(\mathbf{x}_{i,k}, \mathbf{u}_{i,k})$ is an approximation of the dynamics in Eq. (3) by an explicit integration scheme. Hence, the state $\mathbf{x}_i(t_k) := \mathbf{x}_{i,k}$ of agent $i$ is obtained by integrating the dynamics based on the controls $\mathbf{u}_i$ and $\mathbf{x}_{i,k}$ is a function of the initial agent state $\mathbf{x}_{i,0}$ and the strategy $\mathbf{u}_i \in \mathbb{R}^{n_{u,i} \times K}$.

**EPO Optimization Problem.** The solution of the resulting discrete-time OCP is given by

$$\mathbf{u}^* = \arg\min_{\mathbf{u}} \sum_{k=0}^{K-1} p(\mathbf{x}_k, \mathbf{u}_k) + \bar{s}(\mathbf{x}_K), \qquad (5)$$

with discrete-time joint state $\mathbf{x}_k \in \mathbb{R}^{n_x}$, control $\mathbf{u}_k \in \mathbb{R}^{n_u}$ at timestep $k$ and joint strategy $\mathbf{u} \in \mathbb{R}^{n_u \times K}$. Assuming inference of the game parameters $\mathbf{p} = \phi_\theta(\mathbf{o})$ based on observations $\mathbf{o}$, and interpreting the cost as an energy function, similar to [9], the cost terms are now functions of the observations depending on some learnable parameters $\theta$. Let, $E_{\mathbf{p},i}^{\mathrm{own}}(\cdot)$ be the agent-specific energy,

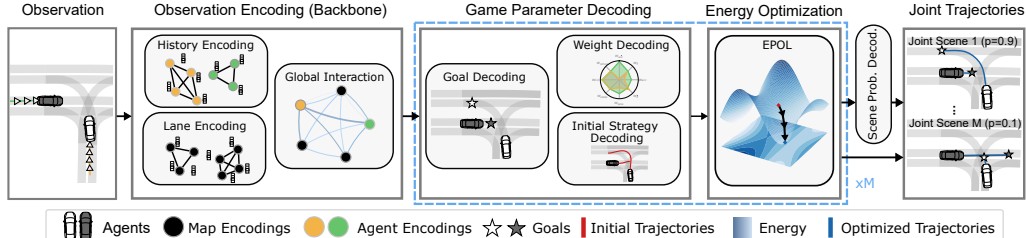

Figure 2: System Architecture. First, a vectorized observation representation is encoded. To handle multi-modality, the decoders predict context-dependent goal positions, initial strategies, and weights for energy parameterization. Parallel optimization problems are solved, resulting in $M$ joint strategies, whereas the dotted blue box represents the parallelization of modes. Unrolling the dynamics generates scene-consistent [26] joint trajectories. Further, a probability decoder predicts scene probabilities.

which can contain running and terminal costs, and $E_{\mathbf{p},i,j}^{\text{pair}}(\cdot)$ the pairwise interaction energy, whereas both are summed over all $K$ timesteps. Combining Eq. (4) and (5) leads to the energy optimization

$$\mathbf{u}^* = \arg\min_{\mathbf{u}} \sum_{i=1}^{N} E_{\mathbf{p},i}^{\text{own}}(\mathbf{u}_i, \mathbf{o}) + \sum_{1 \leq i < j}^{N} E_{\mathbf{p},i,j}^{\text{pair}}(\mathbf{u}_i, \mathbf{u}_j, \mathbf{o}). \tag{6}$$

The energy interpretation, now allows to apply EBM learning techniques.

*SDV Running Example: Unknown game parameters in SDV applications can be the drivers' goals and individual preferences for reaching these goals compared to minimizing control efforts. However, we can use the historical trajectories of traffic participants and maps to infer these parameters.*

## 4 Practical Implementation

**Problem Formulation.** We assume access to an object-based representation of the world consisting of agent histories and (optional) map information as visualized for an SDV example in Fig. 2. Let an observation $\mathbf{o} = (\mathbf{h}, \mathbf{m})$ be defined by a sequence of *all agents* historic 2-D positions $(x, y)$, denoted by $\mathbf{h}$, with length $H$, and by an optional high-definition map $\mathbf{m}$. Our goal is to predict multi-modal future joint strategies $\mathbf{U} \in \mathbb{R}^{n_u \times K \times M}$ and the associate scene-consistent [26] future joint states $\mathbf{X} \in \mathbb{R}^{n_x \times K \times M}$ of all agents and probabilities $\boldsymbol{\pi} \in [0,1]^M$ for all $M$ joint futures.

Let $\mathbf{U}$ represent $M$ different joint strategies $\mathbf{u}^m \in \mathbb{R}^{n_u \times K}$ with modes $m = 1, \ldots, M$, obtained by $M$ *parallel energy minimizations* (Eq. 6). As minimization with gradient-based solvers can induce problems with local optima, we propose to learn initial strategies $\mathbf{U}^{\text{init}} \in \mathbb{R}^{n_u \times K \times M}$ with a neural network consisting of $M$ strategies $\mathbf{u}^{\text{init},m} \in \mathbb{R}^{n_u \times K}$. In addition, for every mode, the network predicts parameters $\mathbf{p}^m \in \mathbb{R}^{n_p}$, whereas $\mathbf{P} \in \mathbb{R}^{n_p \times M}$ describes the parameters of all modes. Concretely, $\mathbf{p}^m$ contains the weights $\mathbf{w}^m \in \mathbb{R}^{n_w}$ and goals $\mathbf{g}^m \in \mathbb{R}^{2 \times N}$ of all agents. Hence, our goal is to estimate a conditional mixture distribution $p(\mathbf{X}|\mathbf{o}) = \sum_{m=1}^{M} \pi^m(\mathbf{o}) \, p(\mathbf{x}^m|\mathbf{o})$, with mixture weights (probabilities for each mode) given by $\pi^m(\mathbf{o})$. The energy minimization and unrolled dynamics define $p(\mathbf{x}^m|\mathbf{o})$. A detailed probabilistic formulation is given in App. D.

### 4.1 Observation Encoding and Decoding of Game Parameters and Probabilities

**Observation Encoding.** Given observations $\mathbf{o}$, the first step is to encode agent-to-agent and agent-to-lane interactions. Inspired by [22], we use different hierarchical graph neural network *backbones* for observation encoding. We first construct polylines $\mathcal{P}$ based on a vectorized environment representation of the agent histories and map elements. The resulting subgraphs are encoded with separate *agent history* $\phi^{\text{hist}}$ and *lane encoders* $\phi^{\text{lane}}$, followed by a high-level global interactions graph. Sec. 5 and App. F.1 provide additional information. Let $\mathbf{z}_i$ be the updated feature of agent $i$ extracted from $\mathbf{z}$ resulting after the global interaction graph. Next, we will describe the different decoders of the game parameters $\mathbf{P}$ and initial strategies $\mathbf{U}_{\text{init}}$ which are all implemented by MLPs.

**Goal Decoding.** As [40], we employ a *goal decoder* to model $p_i^{\text{goal}}(\mathbf{g}_i|\mathbf{o})$ with a categorical distribution over $G$ 2-D goal locations, capturing agent intents (e.g., lane keeping vs. lane changing). We extract multiple goal positions per agent $\mathbf{G}_i \in \mathbb{R}^{2 \times M}$ by selecting the top $M$ goals from $p_i^{\text{goal}}(\cdot)$.

**Weight Decoding.** The energy structure (7) follows a linear combination of features with multi-modal weights $\mathbf{W} \in \mathbb{R}^{n_w \times M}$, which is the concatenation of all agents time-invariant self-dependent weights $\mathbf{W}_i^{\text{own}}$ and pairwise weights $\mathbf{W}^{\text{pair}}$. Two weight decoders predict these: The *agent weight decoder* $\mathbf{W}_i^{\text{own}} = \phi^{\text{own}}(\mathbf{z}_i)$ predicts the weights $\mathbf{W}_i^{\text{own}}$, based on the agent features $\mathbf{z}_i$ from a *single* agent $i$. The *interaction weight decoder* $\mathbf{W}^{\text{pair}} = \phi^{\text{pair}}(\mathbf{z}^{\text{all}})$ predicts all pairwise weights $\mathbf{W}^{\text{pair}}$ at once based on the input $\mathbf{z}^{\text{all}}$, which is the concatenation of the agent features $\mathbf{z}_i$ from *all* $N$ agents.

**Initial Strategy Decoding.** It is important to note that gradient-based methods may not always converge to global or local optima. However, these methods can be highly effective when the solver is initialized close to an optimum [41]. Hence, we learn $M$ initial joint strategies, denoted by $\mathbf{U}^{\text{init}}$. More concretely, the *initial strategy decoder* predicts $\mathbf{U}_i^{\text{init}} = \phi^{\text{init}}(\mathbf{z}_i)$. The parameters of the goal, agent weight, and strategy decoders are shared for all agents. Hence, the computation is parallelized.

**Scene Probability Decoding.** The $M$ optimizations yield joint strategies $\mathbf{U}$. Unrolling the dynamics yields $M$ future joint state trajectories $\mathbf{X}$. The *scene probability decoder* estimates probabilities for each future $\boldsymbol{\pi} = \phi^{\text{prob}}(\mathbf{z}^{\text{prob}})$ using the concatenation of $\mathbf{z}^{\text{all}}$ and $\mathbf{X}$ as input, denoted by $\mathbf{z}^{\text{prob}}$.

### 4.2 Energy Optimization

**Energy Structure.** Let $\mathbf{w}_i^{\text{own},m}$, $\mathbf{w}_{i,j}^{\text{pair},m}$, and $\mathbf{u}_i^m \in \mathbb{R}^{n_u \times K}$ describe the weight vectors and strategies of mode $m$ and agent $i$. The EPOL solves the $M$ optimization problems *in parallel* based on the predicted $\mathbf{P}$ and $\mathbf{U}^{\text{init}}$, whereas the energies in Eq. (6) have the structure of a linear combination of weighted nonlinear vector-valued functions $c(\cdot)$ and $d(\cdot)$ given by

$$E_{\mathbf{p},i}^{\text{own}}(\mathbf{u}_i^m, \mathbf{o}) = \frac{1}{2} \left\| (\mathbf{w}_i^{\text{own},m})^\intercal c(\mathbf{u}_i^m, \mathbf{g}_i^m) \right\|^2, \ \ E_{\mathbf{p},i,j}^{\text{pair}}(\mathbf{u}_i^m, \mathbf{u}_j^m, \mathbf{o}) = \frac{1}{2} \left\| (\mathbf{w}_{i,j}^{\text{pair},m})^\intercal d(\mathbf{u}_i^m, \mathbf{u}_j^m) \right\|^2,$$
(7)

which allows incorporating domain knowledge into the training process. $c(\cdot)$ includes agent-specific costs, which, for example, can induce goal-reaching behavior while minimizing control efforts. $d(\cdot)$ is a distance measure between two agent geometries.

**Differentiable Optimization.** The structure of (7) allows us to solve parallel optimizations using the differentiable Nonlinear Least Square solvers of [32]. The implementation uses the second-order Levenberg–Marquardt method [42]. Hence, we minimize Eq. (6) by iteratively taking $S$ steps $\mathbf{u}_{s+1} = \mathbf{u}_s + \alpha \Delta \mathbf{u}$. $s$ describes the iteration index with $s = 1, \ldots, S$ and $\alpha$ is a stepsize $0 < \alpha \leq 1$. $\Delta \mathbf{u}$ is found by linearizing the energy around the current joint strategy $\mathbf{u}$ and subsequently solving a linear system [32]. During training, we can then backpropagate gradients through the unrolled *inner loop energy minimization* based on a loss function of the *outer loop loss minimization*.

### 4.3 Training Objectives and Transfer to Control

**Training Objectives.** We follow prior work [23, 40] and minimize the multi-task loss $\mathcal{L} = \lambda_1 \mathcal{L}^{\text{imit}} + \lambda_2 \mathcal{L}^{\text{goal}} + \lambda_3 \mathcal{L}^{\text{prob}}$, with scaling factors $\lambda_1, \lambda_2, \lambda_3$. The imitation loss $\mathcal{L}^{\text{imit}}$ is a distance of the joint future closest to the ground truth. $\mathcal{L}^{\text{goal}}$ computes the negative log-likelihood (NLL) using the predicted goals $\mathbf{G}$ and $\mathcal{L}^{\text{prob}}$ the NLL for the future joint states $\mathbf{X}$. App. F.4 provides further details.

**Transfer to Single-Agent Control.** Assuming a controllable SDV with index $i = 0$ in the scene, we adopt a similar approach as [43]. We extract the most likely mode index $m^* = \arg\max_m \pi^m$ and the SDV's strategy $\mathbf{u}_{i=0}^{m^*}$. Using model-predictive control (MPC), we can execute the first control vector $\mathbf{u}_{i=0,k=0}^{m^*}$ or rely on an underlying tracking controller.

## 5 Experimental Evaluation

The experiments investigate the following research questions: *Q1*: Is the approach applicable to different motion forecasting backbones, and does it enhance the predictive performance? *Q2*: Can the method predict multi-modal joint futures in an interpretable manner?

**Evaluation Environments.** *RPI* is a dataset of simulated multi-modal mobile robot pedestrian interaction (RPI) constructed based on the implementation of [43]. *exiD* is a real-world dataset of interactive scenarios captured by drones at different highway-ramp locations [44]. *Waymo Interactive* is a large-scale urban dataset [20]. The *CARLA* simulator [45] is used to construct a dataset and

closed-loop simulations of left-turn scenarios. The datasets contain 60,338 (RPI), 290,735 (exiD), 176,583 (Waymo) and 2,959 (CARLA) samples, respectively. Details are given in App. E.

**Metrics.** We employ standard prediction metrics [46, 23, 35]. The *minADE* calculates the $L_2$ norm of a *single-agent trajectory* out of $M$ predictions with the minimal distance to the groundtruth. The *minFDE* is similar to the minADE but only evaluated at the last timestep. The *minSADE* and *minSFDE* are the scene-level equivalents to minADE and minFDE, calculating the $L_2$ norm between *joint trajectories* and joint ground truth [26]. We further calculate the overlap rate *OR* as [35] of the most likely-joint prediction, which measures the scene consistency. When using marginal prediction methods, joint metrics (minSADE, minSFDE, OR) are computed by first ordering the single agent predictions according to their marginal probabilities and constructing a joint scenario accordingly.

**Baselines.** *Constant Velocity* (CV) is a kinematic baseline, achieving good results for predicting pedestrians [47] or highway vehicles [9]. We also use the following SOTA architectures as baselines and observation encoding backbones: *V-LSTM* [20], *HiVT-M*, a modified version of [48], and the unpublished model *VIBES*. For details, refer to App. F. These baselines employ a marginal loss formulation, minimizing the minADE for trajectory regression and using a classification loss similar to [40] to estimate probabilities. However, this formulation of a *marginal distribution per actor* may lead to inconsistencies in future trajectories. For a fair comparison, we introduce `Backbone+SC` (SC: scene control prediction) baselines that predict control values like [49] and minimize a scene-consistent loss, consisting of minSADE and the same scene probability loss as our approach ($\mathcal{L} = \mathcal{L}^{\text{imit}} + \mathcal{L}^{\text{prob}}$ from Sec. 4.3). These baselines also approximate a *joint distribution* over future states *per scene*. We further compare to TGL [2], another differentiable (convex) optimization-based approach (Tab. 1).

**Energy Features and Dynamics.** We use an explicit Euler-forward integration scheme of dynamics $f_i(\cdot)$ (dynamically-extended unicycles [38]). Agents' geometries are approximated by a circle of radius $r_i$. Hence, $d(\cdot)$ in Eq. (7) is a point-to-point distance, active when the circles overlap. Depending on the environment, specific features in $c(\cdot)$ penalize deviations from a goal location, high controls and control derivations, velocities, differences to a reference path or velocity, as well as violations of state and control bounds. App. F.3 provides additional information.

## 5.1 Results

**Quantitative and Qualitative Evaluation.** Tab. 2 and 3 summarize the results of different models on the exiD and RPI datasets. In App. G.4, we present a statistical analysis for the models on exiD and Waymo with various random seeds. On average, our implementation, consistently outperforms the baselines in all joint distance-based metrics (minSADE, minSFDE) across all backbones due to the game-theoretic inductive bias, with a greater significance on the more diverse Waymo dataset. Especially joint metrics are important, as they measure the scene consistency, which is also underlined by [50]. To answer Q1: Our approach can be applied to different backbones and improves predictions in joint distance-based metrics.

Table 2: Predictive performance comparison on the exiD test dataset. ADE, FDE, SADE, and FDE are computed as the minimum over $M = 5$ predictions in [m]. **Bold** marks the best result and underlined the second best of each group of approaches, which uses the same observation encoding backbone. Lower is better.

| Method | Marginal ↓ | | Joint ↓ | | |
| | ADE | FDE | SADE | SFDE | OR |
|---|---|---|---|---|---|
| V-LSTM | 1.25 | 3.64 | 1.98 | 3.90 | 0.031 |
| + SC | 0.82 | 1.95 | 1.07 | 2.63 | 0.010 |
| + TGL | 0.96 | 2.32 | 1.16 | 2.73 | 0.048 |
| + Ours | **0.80** | **1.89** | **0.99** | **2.37** | **0.008** |
| VIBES | 1.35 | **1.68** | 1.93 | 2.93 | 0.021 |
| + SC | **0.73** | 1.71 | 1.01 | 2.47 | 0.007 |
| + Ours | 0.83 | 1.99 | **0.99** | **2.40** | **0.006** |
| HiVT-M | 1.83 | 2.02 | 2.39 | 2.86 | 0.013 |
| + SC | **0.78** | **1.90** | 1.04 | 2.57 | 0.008 |
| + Ours | 0.83 | 1.97 | **1.03** | **2.48** | **0.007** |
| CV | 1.16 | 2.87 | 1.16 | 2.87 | 0.007 |

In Fig. 3 (a) our approach predicts the correct joint future (mode 2) with a high mode probability and also outputs another reasonable alternative future. To answer Q2, consider the definition of interpretability in the context of SDV of [12]. Our approach allows us to visualize intermediate latent representations (feature weights). For example, the weight for reaching the reference velocity $w_{\text{vref}}$ is higher in mode 1 for the blue vehicle and lower

for the red vehicle. That provides insights into the decision-making process (App. G.2). Fig. 3 (b) and (c) present a closed-loop result in another scenario and open-loop predictions during a highway merge. Additional results can be found in App. G.

**Ablation Study.** This section ablates the influence of learning an initialization and using goal-related features as illustrated in Tab. 4. Turning off goal features inhibits goal-reaching behavior, which is important for modeling human behavior [37]. Hence, the performance declines in all metrics. When the learned initialization is turned off, and agents' controls are initialized with zeros, we observe a decline in performance. These findings highlight the importance of the algorithmic components.

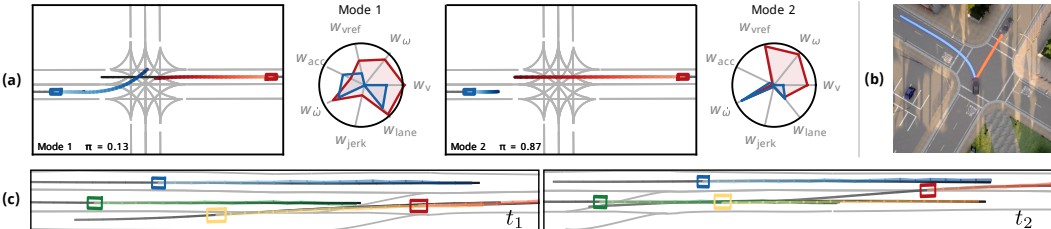

Figure 3: Qualitative joint predictions. (a) Open-loop (CARLA) for two modes and self-dependent weights $\mathbf{W}_i^{\text{own}}$ normalized w.r.t. the maximum weight in the sequence. Hence, weights of the same feature are comparable between the two modes, but weights of different features are not. The right mode is predicted with high probability. (b) Closed-loop control (CARLA) of the SDV (blue) in another scenario, where the SDV accurately anticipates that the other vehicle will yield. (c) Open-loop (exiD) for two time steps during a highway merge (lane change of yellow and red). Our model accurately predicts the resulting interactions and produces scene-consistent predictions. (b) and (c) illustrate the most likely joint trajectories. The ground truth is visualized with colors from dark grey to black and the map in light grey lines. Agent (rectangle) trajectories have different colors.

Table 3: Predictive performance of different methods on the RPI test dataset. The metrics and formatting are the same as in Tab. 2, but $M = 2$.

| | Marginal ↓ | | Joint ↓ | | |
|---|---|---|---|---|---|
| Method | ADE | FDE | SADE | SFDE | OR |
| V-LSTM | 0.12 | 0.20 | 0.13 | 0.23 | 0.006 |
| + SC | 0.04 | 0.11 | 0.04 | 0.11 | 0.001 |
| + Ours | **0.03** | **0.08** | **0.03** | **0.08** | **0.000** |

Table 4: Ablation study investigating the impact of learned initialization, and goal-related features (exiD test set, VIBES backbone).

| | Marginal ↓ | | Joint ↓ | | |
|---|---|---|---|---|---|
| Method | ADE | FDE | SADE | SFDE | OR |
| Ours (full) | **0.83** | **1.99** | **0.99** | **2.40** | **0.006** |
| No init | 0.89 | 2.04 | 1.01 | 2.47 | 0.007 |
| No goal | 0.89 | 2.13 | 1.10 | 2.64 | 0.009 |

## 5.2 Limitations

As commonly reported in the literature, game-theoretic motion planning suffers from increased runtime, especially with an increasing number of agents. While our implementation scales well with the number of modes (nearly constant runtime) due to parallelization (see App. G.4)), that effect is also present in our non-runtime optimized application. Future work could apply decentralized optimizations similar to [51] to reduce the runtime. Our implementation is limited by a fixed number of agents due to the requirement of fixed-size optimization variables of [32]. Future work should dynamically identify interacting agents, considering that not all agents are constantly engaged in a scene. That could be done utilizing the already existing attention mechanisms, similar to [52], or heuristics, similar to [35]. The proposed EPO further opens opportunities for various future algorithms, combining different types of energy structures, optimization, and differentiation techniques, which could also overcome scalability limitations induced by using energy features in our application. Lastly, the assumption of pairwise equal preferences for collision avoidance is restrictive.

## 6 Conclusions

This work connected differential games, optimal control, and EBMs. Based on these findings, we developed a practical implementation that improves the performance in joint distance metrics of various neural networks in scene-consistent motion forecasting experiments and applies to motion control. Similar to [53], we hope that by highlighting the connection between these fields, researchers in these three communities will be able to recognize and utilize transferable concepts across domains.

**Acknowledgments and Disclosure of Funding** This work was supported by the Federal Ministry for Economic Affairs and Climate Action on the basis of a decision by the German Bundestag and the European Union in the Project KISSaF - AI-based Situation Interpretation for Automated Driving. The authors would like to thank Stefan Schütte for fruitful discussions.

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

# A List of Abbreviations

| Abbreviation | Description |
|---|---|
| App. | Appendix |
| ADE | average displacement error |
| CDE | conditional density estimation |
| CV | Constant Velocity |
| EBM | Energy-based Model |
| EPOL | Energy-based Potential Game Layer |
| EPO | Energy-based Potential Game |
| Eq. | Equation |
| FDE | final displacement error |
| Fig. | Figure |
| GB | gradient-based |
| HiVT-M | Hierarchical Vector Transformer Modified |
| IFT | implicit function theorem |
| LB | learning-based |
| LSTM | long short-term memory |
| MLE | maximum likelihood estimation |
| MLP | multilayer perceptron |
| MB | model-based |
| NC | noise contrastive |
| NE | Nash equilibrium |
| NN | neural network |
| NLL | negative log likelihood |
| Nonlin. | nonlinear |
| Num. | Number |
| OCP | optimal control problem |
| OLNE | open-loop Nash equilibrium |
| OR | overlap rate |
| PDG | potential differential game |
| RPI | robot pedestrian interaction |
| SADE | scene average displacement error |
| SB | sampling-based |
| SC | scene control prediction |
| Sec. | Section |
| Tab. | Table |
| SDV | self-driving vehicle |
| SADE | scene average displacement error |
| SFDE | scene final displacement error |
| SOTA | state-of-the-art |
| UN | unrolling |
| VIBES | Vectorized Interaction-based Scene Prediction |
| V-LSTM | Vector-LSTM |

# B Extended Related Work

**Data-driven Motion Forecasting.** Deep learning motion forecasting approaches use different observation inputs. [49] uses birds-eye-view images, which have a high memory demand and can lead to discretization errors. [22] propose to use a vectorized environment representation instead, and [46] uses raw-sensor data. Many approaches utilize an encoder-decoder structure with convolutional neural networks [24], transformers [23], or graph neural networks [26] to model multi-agent interactions. In addition to deterministic models [23], various generative models, such as Generative Adversarial Networks (GANs) [54] and Conditional Variational Autoencoder formulations [10], as well as Diffusion Models [55], are used to produce multi-modal predictions. Predicting goals in hierarchical approaches like [40] can further increase the predictive performance using domain knowledge from

the map. The prediction can also be conditioned on the future trajectory [10] of one agent. However, these conditional forecasts might lead to overly confident anticipation of how that agent may influence the predicted agents [56]. To include domain knowledge such as system dynamics into the learning process, it is also common practice [11, 49, 10] to forecast the future control values of all agents and then to unroll a dynamics model produce the future states.

**Differentiable Optimization for Motion Planning.** Differentiable optimization has also been applied in motion planning for SDVs. Some works, [57] and [34] impose safety constraints using differentiable control barrier functions or gradient-based optimization techniques in static environments. The authors of [33] and [58] couple a differentiable single-agent motion planning module with learning-based motion forecasting modules. In contrast, our work performs multi-agent joint optimizations in parallel, derived from a game-theoretic potential game formulation. Game-theoretic formulations can overcome overly conservative behavior when used for closed-loop control [4].

**Model-based Reinforcement Learning.** Model-based reinforcement learning techniques have also been applied for decision-making in interactive environments. Schwarting et al. [59] present an approach for *online* model-based multi-agent reinforcement learning evaluated in a two-player racing simulation. In contrast, to our work, [59] also learns the dynamics through a multi-agent world model, which is further used for learning a policy through imaged self-play.The approach of [60] follows a similar path, estimating upper confidence bounds on agents' value functions, which are then utilized to identify coarse-correlated equilibria [61] (CEE). The concept of CEE introduces a broader perspective of the Nash equilibria employed in this study. The approach is assessed in an SDV simulator. The downsides of both approaches are two-fold: First, they assume knowledge about the reward/ cost function of all agents, which is unpractical in many real-world applications. For instance, the intentions of human drivers, such as their goals, are not directly observable. Second, [59] and [60] are *online* approaches, learning by interacting with the environment, which is again unpractical in many safety-critical robotics applications. [62] and it's extended version [6], introduce a single-agent model-based *offline* reinforcement learning approach, using a stochastic action-conditioned world/ dynamics model to predict real-world interactions. However, learning action-conditioned world models can lead to overly-confident predictions of how that agent may influence others, as indicated in the study of [56]. In contrast, our *offline* approach models interactions through learned costs/rewards with given dynamics. We can interpret cost learning as a type of multi-agent inverse reinforcement learning, also called inverse game learning [7]. The forward pass can be interpreted as solving a multi-agent model-based reinforcement learning problem, utilizing planning with learned cost [63].

**Energy-based Models.** The work of Xie et al. [64] proposes parametrizing an EBM with a neural network, and learning is performed with type (i) MLE. [65] uses an EBM for model-based single-agent planning. Our work is also related to the cooperative training paradigm [66, 67, 68, 69], in which a (fast-thinking) latent variable model and a (slow-thinking) EBM are trained together. These works parameterize the latent variable model used initializing the EBM, with a generator [67], a variational auto-encoder [68], or a normalizing flow [66]. In contrast, our work does not separate the training of the initialization and the EBM with different networks.

## C   Theorems

This section provides the full theorem of [18]:

**Theorem C.1.** *For a differential game* $\Gamma_{\mathbf{x}_0}^T := \left( T, \{\mathbf{u}_i\}_{i=1}^N, \{C_i\}_{i=1}^N, f \right)$*, if for each agent* $i$*, the running and terminal costs have the following structure* $L_i(\mathbf{x}(t), \mathbf{u}(t), t) = p(\mathbf{x}(t), \mathbf{u}(t), t) + c_i(\mathbf{x}_{-i}(t), \mathbf{u}_{-i}(t), t)$ *and*

$$S_i(\mathbf{x}(T)) = \bar{s}(\mathbf{x}(T)) + s_i(\mathbf{x}_{-i}(T)),$$

*then, the open-loop control input $\mathbf{u}^* = (\mathbf{u}_1^*, \cdots, \mathbf{u}_N^*)$ that minimizes the following*

$$\min_{u(\cdot)} \int_0^T p(\mathbf{x}(t), \mathbf{u}(t), t)dt + \bar{s}(\mathbf{x}(T))$$
$$s.t. \ \dot{x}_i(t) = f_i\left(\mathbf{x}_i(t), \mathbf{u}_i(t), t\right),$$

*is an OLNE of the differential game $\Gamma_{\mathbf{x}_0}^T$, i.e., $\Gamma_{x_0}^T$ is a potential differential game.*

Proof: See [18], with original proof provided by [8].

Here besides the potential functions $p$ and $\bar{s}$, $s_i$ and $c_i$ are terms that are required to not depend on the state or control of agent $i$.

## D Extended Problem Formulation

This section provides an extended description of the problem formulation (see Sec. 4. The goal is to estimate a conditional probability distribution $p(\mathbf{X}|\mathbf{o})$, whereas $\mathbf{o}$ describes the observed context (random variable), and $\mathbf{X}$ the random variable of all agents' future joint state trajectories. To make the approach tractable, we marginalize over agents' joint intents, to get $M$ modes of future joint strategies $p(\mathbf{X}|\mathbf{o}) = \sum_{m=1}^M \pi^m(\mathbf{o}) \, p(\mathbf{x}^m|\mathbf{o})$, which describes a mixture distribution with mixture weights given by $\pi^m(\mathbf{o})$. Here, similar to prior work [11], we model uncertainty over the discrete joint modes $M$ with a softmax distribution $\pi^m(\mathbf{o}) = \frac{\exp(f_m(\mathbf{o}))}{\sum_j \exp(f_j(\mathbf{o}))}$, whereas $f_m(\mathbf{o})$ is the ouput of a neural network. We receive $p(\mathbf{x}^m|\mathbf{o})$ as follows: First we model a distribution $p(\mathbf{u}^m|\mathbf{o})$. Here, $\mathbf{u}^m$ is received by optimizing the energy function containing parameters predicted by the neural network based on observation $\mathbf{o}$. Moreover, the joint strategy $\mathbf{u}^m$ and the joint future state trajectory $\mathbf{x}^m$ are connected by a determinstic function $\mathbf{x}^m = f_{\text{unroll}}(\mathbf{x_0}, \mathbf{u}^m)$, which unrolls the deterministic system dynamics from the current (measured) state $\mathbf{x_0}$ using the predicted strategy $\mathbf{u}^m$. That could be written as the following dirac distribution $p(\mathbf{x}^m|\mathbf{u}^m, \mathbf{o}) = \delta\left(\mathbf{x}^m - f_{\text{unroll}}(\mathbf{x_0}, \mathbf{u}^m)\right)$ and hence using the law of probability $p(\mathbf{x^m}|\mathbf{o}) = \int p(\mathbf{x}^m|\mathbf{u}^m, \mathbf{o}) \cdot p(\mathbf{u^m}|\mathbf{o}) \, d\mathbf{u}^m$. By plugging together the relationships, we receive:

$$p(\mathbf{X}|\mathbf{o}) = \sum_{m=1}^M \pi^m(\mathbf{o}) \, p(\mathbf{x}^m|\mathbf{o})$$

$$= \sum_{m=1}^M \pi^m(\mathbf{o}) \int p(\mathbf{x}^m|\mathbf{u}^m, \mathbf{o}) \cdot p(\mathbf{u}^m|\mathbf{o}) \, d\mathbf{u}^m \quad (8)$$

$$= \sum_{m=1}^M \pi^m(\mathbf{o}) \int \delta(\mathbf{x}^m - f_{\text{unroll}}(\mathbf{x_0}, \mathbf{u}^m)) \cdot p(\mathbf{u}^m|\mathbf{o}) \, d\mathbf{u}^m.$$

## E Datasets

### E.1 RPI

The RPI dataset is a synthetic dataset of simulated mobile robot pedestrian interactions. Multi-modal demonstrations are generated by approximately solving a two-player differential game ($N = 2$) with the iterative linear-quadratic game implementation of [17, 43] based on different start and goal configurations. Fig. 4 illustrates the dataset construction. The robot's initial positions (white circle) and goal locations (white star) are the same in all solved games. In contrast, the initial state (dark grey circle) and goal location (dark grey stars) of the pedestrian move on a circle, as illustrated on the left graphic in Fig. 4. The agents are tasked to reach a goal location given an initial start state while avoiding collisions and minimizing control efforts. As solving the game once leads to a uni-modal local strategy, this work follows the implementation of Peters et al. [43]. It solves the game for a given initial configuration multiple times based on different sampled strategy initializations. Afterward, the resulting strategies are clustered. The clustered strategies represent multi-modal strategies of the *main*

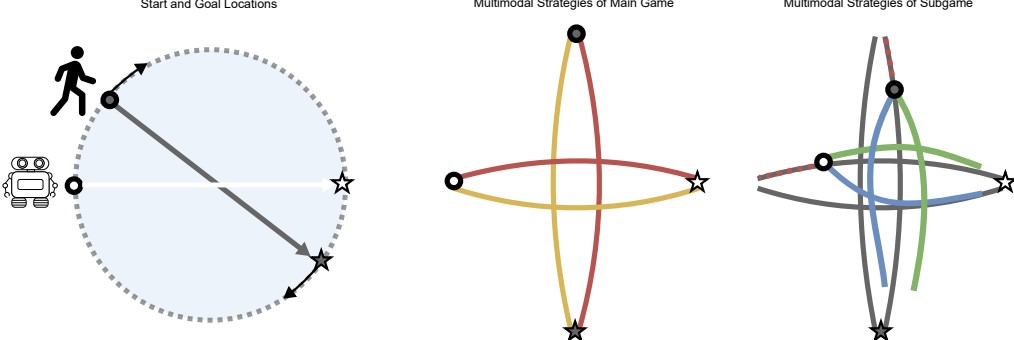

Figure 4: Dataset construction for RPI dataset. Left: First initial and goal states and game parameters are sampled. Middle: A *main game* is solved multiple times based on the sampled game configuration with subsequent result clustering. That leads to multimodal strategies (red and yellow). The agent moves according to the multimodal strategies of the main game. After a time step $\Delta t$, a *sub game* is solved. The results are multimodal strategies (blue and green) of the subgame. The histories (dotted red) and multi-modal strategies of the sub game build a demonstration for training and evaluation.

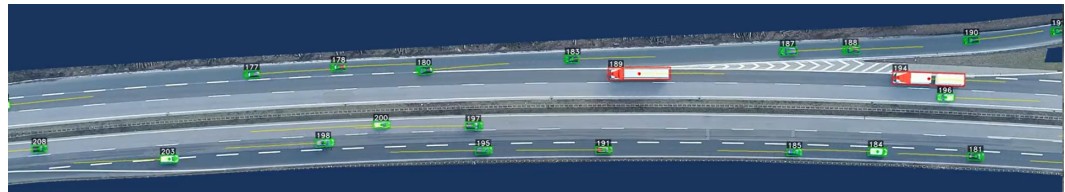

Figure 5: An exemplary highly-interactive scenario from the exiD dataset.

*game*, and they are visualized in red and yellow in Fig. 4. The agents then execute the open-loop controls of the main game's initial strategies. After every time interval $\Delta t = 0.1$, the procedure of game-solving and clustering the results is repeated as long as the agents pass each other. The resulting strategies of the so-called *subgames* are visualized in green and blue on the right of Fig. 4. Based on the history (dotted red line) and the strategies of the subgame (blue and green), we then build a multi-modal demonstration for the dataset. Note that the main game and the corresponding subgames use the same cost function parametrizations, but the agents' preferences for collision avoidance differ between main games.

The resulting dataset is based on 20 main games and their corresponding subgame solutions. Here we draw collision cost parameters from a uniform distribution to enhance demonstration diversity. The resulting dataset contains 60338 samples, whereas we use 47822 ($\sim$80%) for training, 6228 for validation ($\sim$10%), and 6228 ($\sim$10%) for testing. The test set is constructed based on unseen main game configurations. The goal is to predict $M = 2$ joint futures of $T = 4\,\mathrm{s}$ based on a history of $H = 1.8\,\mathrm{s}$ with a time interval of $\Delta t = 0.1$.

### E.2 exiD

The exiD [44] dataset contains $19\,\mathrm{h}$ of real-world highly interactive highway data. Interactions between different types of vehicle classes are rich because the data was recorded by drones flying over seven locations of German highway entries and exits. Highway entries and exits, designed with acceleration and deceleration lanes and high-speed limits, promote interactive lane changes due to high relative speeds between on-ramping and remaining road users. In addition, the most common cloverleaf interchange in Germany requires simultaneous observation of several other road users and gaps between them for safe entry or exit in a short time frame [44].

To further increase the interactivity, this work extracts scenarios with $N = 4$ agents in which at least one agent performs a lane change. The recordings are then sampled with a frequency of $\Delta t = 0.2\,\mathrm{s}$. The different networks are tasked to predict $M = 5$ joint futures of length $T = 4\,\mathrm{s}$ based on a history of $H = 1.8\,\mathrm{s}$. The resulting dataset contains 290735 samples, whereas we use 206592 ($\sim$72%) for training, 48745 for validation ($\sim$16%), and 35398 ($\sim$12%) for testing. To investigate

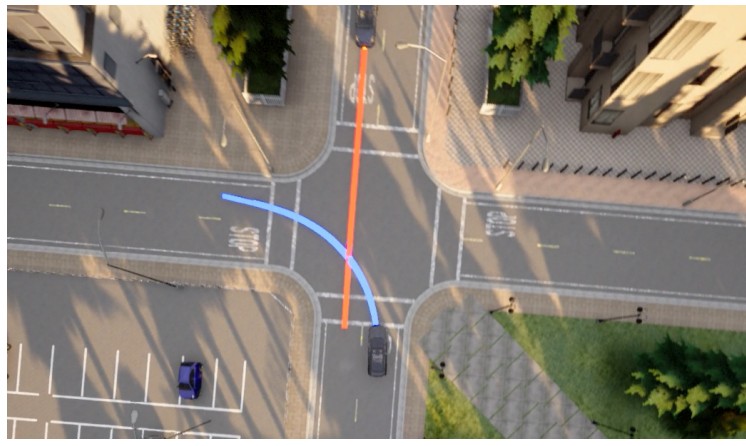

Figure 6: The CARLA left-turning scenario with planned joint trajectories (orange and blue lines) of two agents at an intersection.

the generalization capabilities of the different models, the test set contains *unseen scenarios from a different map* (map 0) than the training and validation scenarios. An exemplary scenario is visualized in Fig. 5.

### E.3 CARLA

The experiments in the CARLA simulator (Version 9.11) use the implementation of [70] to construct interactive scenarios, whereas the SDV is tasked to perform an unprotected-left turn with another vehicle approaching the intersection ($N = 2$) as visualized in Fig. 6. To generate multi-modal demonstrations for training, the agents follow hand-crafted policies to generate different outcomes. The SDV first decides whether to enter the intersection. The other vehicle decides to yield or allow the SDV to pass. Subsequently, the SDV re-evaluates the initial maneuver. This results in different interaction outcomes leading to multi-modal demonstrations.

This work uses four different intersections in Town 04. The approach is tasked to predict $M = 2$ joint futures of length $T = 6$ s based on a history of $H = 1.8$ s. We only use five episodes per intersection of different interaction outcomes for training. In contrast to the exiD and RPI experiments, we use an agent-centric coordinate system with the SDV as the origin. To increase the training dataset size, we perform the following data augmentations: 1) We add additional samples based on the original samples where all observations (history and map) are randomly rotated with a rotation angle drawn from a uniform distribution $\mathcal{U}[-\pi/8, \pi/8]$. In these samples, we 2) add Gaussian noise $\mathcal{N}(0, 0.02)$ to the 2-D positions in the histories. Note that these augmentations are only performed for samples of the training dataset. The resulting dataset contains 2959 samples, whereas we use 1836 ($\sim$62%) for training, 361 ($\sim$12%) for validation, and 762 ($\sim$26%) for testing. The training (intersection 1), validation (intersection 2), and test set (intersection 3 and 4) all differ in terms of the used intersection. During closed-loop control, we evaluate on intersection 3.

In the closed-loop control experiment, the SDV follows the procedure described in Sec. 4 of the main paper to predict the SDV strategy $\mathbf{u}_{i=0}^{m^*}$ and the corresponding state trajectory $\mathbf{x}_{i=0}^{m^*}$. Two PID-controller are used for trajectory tracking. They compute steering angle, braking, or throttle signals to control the SDV in the simulator.

### E.4 Waymo Interactive

The Waymo dataset [20] contains $570$ h hours of real-world driving across six cities in the United States, which were mined for interesting scenarios. We use the *interactive* split of the Waymo dataset, which provides labels for $N = 2$ interacting agents. Moreover, we filter the dataset only to consider vehicle-to-vehicle interactions, whereas we use samples from the official training split for our new training set and samples from the official validation split for testing.

The models are tasked to predict $M = 6$ joint futures of length $T = 8\,\mathrm{s}$ based on a history of $H = 1\,\mathrm{s}$ with $\Delta t = 0.1$. Similar to the CARLA experiments, we use an agent-centric coordinate system with one randomly selected vehicle as the origin, but do not perform data augmentations. The resulting dataset contains 176,583 samples, of which we use 145,584 ($\sim$82%) for training and 30,854 ($\sim$18%) for testing.

# F  Implementation Details

This section provides additional information for the used observation encoding backbones and the game parameter decoders. We further provide details for the used dynamics and energy features.

### F.1  Network Architectures (Backbones and Baselines)

**Lane Encoder.** In all experiments, the lane encoders $\phi^{\text{lane}}$ of all backbones use a PointNet [71] like architecture as [22] with three layers and a width of 64. The polylines are constructed based on vectors that contain a 2-D start and 2-D goal position in a fixed-global coordinate system. Agent polylines also include time step information and are processed with different encoders depending on the used backbone.

**Agent History Encoder.** The V-LSTM (Vector-LSTM) [20] and VIBES (Vectorized Interaction-based Scene Prediction) backbones use an LSTM [72] for agent history encoding with depth three and width 64. Our modified HiVT-M (Hierarchical Vector Transformer Modified) [48] implementation uses a transformer [73] for the encoding of each agent individually. Note that this contrasts with the original implementation, where the encoding transformer already models local agent-to-agent and agent-to-lane interactions. We account for that in a modified global interaction graph as listed below. The transformer has a depth of three and a width of 64.

**Global Interaction.** The V-LSTM backbones update the polyline features in the global interaction graph with a single layer of attention [73] as described by [22]. The HiVT-M and VIBES models use a two-stage attention mechanism. First, one layer of self-attention [73] between the map and agent polyline features is applied using the map features as keys. Afterward, another self-attention layer, whereas the keys are the agents' features. That second attention layer models additional agent-to-agent interaction. The global interaction graph has a width of 128. The two-stage attention module is the main difference between the V-LSTM and the VIBES model.

**Game Parameter and Initial Strategy Decoder.** 3-layer MLPs implement the agent weight, goal, and initial strategy decoders with a width of 64.

**Goal Decoder.** The goal decoder in the exiD and CARLA environment follows [40]. It takes as input the concatenation of an agent feature $\mathbf{z}_i$ and $G = 60$ possible goal points, denoted by $\mathbf{z}^{\text{goal}}$. The goal points are extracted from the centerlines of the current and neighboring lanes. If there exists no neighboring lane, we take the lane boundaries. The decoder $\phi^{\text{goal}}$ then predicts the logits of a categorical distribution per agent $\mathbf{l}_i^{\text{goal}} = \phi^{\text{goal}}(\mathbf{z}^{\text{goal}})$. During training and evaluation, the method takes the $M$ most-likely goals $\mathbf{G}_i$ for all modes of agent $i$. Probabilities for the goals per agent are computed by $\mathbf{PR}_i^{\text{goal}} = \text{softmax}(\mathbf{l}_i^{\text{goal}})$. The prediction of goals is made in parallel for all agents. In the Waymo environment the goal decoder uses the same architecture but directly regresses the goals (see Equ. (12)).

**Scene Probability Decoder.** The scene probability decoder uses a 3-layer MLP with width $16 \times M$ and predicts logits $\mathbf{l}^{\text{prob}}$ for the $M$ scene modes. The scene probabilities (mixture weights) are derived by applying the softmax operations $\boldsymbol{\pi} = \text{softmax}(\mathbf{l}^{\text{prob}})$.

The goal, agent weight, and scene probability decoder use batch normalization. The interaction weight decoder, initial strategy decoder, lane encoder, and transformer agent encoder use layer normalization.

In the CARLA experiment we scaled the width of all networks by half to mitigate overfitting. We also experimented with downscaling the network with by a factor of four, but saw no increase in performance for the baselines and our method.

## F.2 Dynamics

Let $x_k$ and $y_k$ denote a 2-D position and $\theta_k$ the heading. In exiD and RPI, the origin of the system is fixed. In CARLA, the origin is the SDV, whereas the $x$ axis is aligned with the SDV. Similarly, in Waymo, the origin is a randomly selected vehicle. $v_k$ is the velocity, $a_k$ the acceleration, $\omega_k$ the turnrate, and $\Delta t$ a time interval. Hence, $n_x = 4 \times N$ and $n_u = 2 \times N$. The discrete-time dynamically-extended unicycle dynamics [38, Chapter 13] are given by:

$$
\begin{aligned}
x_{k+1} &= x_k + v_k \cos\left(\theta_t\right) \Delta t \\
y_{k+1} &= y_k + v_k \sin\left(\theta_t\right) \Delta t \\
v_{k+1} &= v_k + a_k \Delta t \\
\theta_{k+1} &= \theta_k + \omega_k \Delta t
\end{aligned}
\tag{9}
$$

## F.3 Energy Features and Optimization

**Energy Features.** The energy function in the RPI experiment uses the following agent-dependent features: $c(\cdot) = [c_{\text{goal}}, c_{\text{vel}}, c_{\text{acc}}, c_{\text{velb}}, c_{\text{accb}}, c_{\text{turnr}}, c_{\text{accb}}, c_{\text{turnrb}}]$. In the RPI experiments, the goal is given and not predicted. The agent-dependent energy features in the exiD experiments are given by $c(\cdot) = [c_{\text{goal}}, c_{\text{lane}}, c_{\text{vref}}, c_{\text{vel}}, c_{\text{acc}}, c_{\text{jerk}}, c_{\text{turnr}}, c_{\text{turnacc}}]$. The agent-dependent energy features in the CARLA experiments are given by $c(\cdot) = [c_{\text{lane}}, c_{\text{vref}}, c_{\text{vel}}, c_{\text{acc}}, c_{\text{jerk}}, c_{\text{turnr}}, c_{\text{turnacc}}]$ and in the Waymo experiments by $c(\cdot) = [c_{\text{goal}}, c_{\text{vel}}, c_{\text{acc}}, c_{\text{jerk}}, c_{\text{turnr}}, c_{\text{turnacc}}]$. $c_{\text{goal}}$ is a terminal cost penalizing the position difference of the last state to the predicted goal. $c_{\text{lane}}$ minimizes the distance of the state trajectory to the reference lane to which the predicted goal point belongs. Note that different goal points can be predicted for the modes, and as a result, different lanes can be selected to better model multi-modality. $c_{\text{vref}}$ is the difference between the predicted and map-specific velocity limit. The other terms are running cost, evaluated for all timesteps and penalize high velocities ($c_{\text{vel}}$), accelerations ($c_{\text{acc}}$), jerks ($c_{\text{jerk}}$), as well as turn rates ($c_{\text{turnr}}$) and turn accelerations ($c_{\text{turnacc}}$). An index b marks a soft constraint implemented as a quadratic penalty, active when the bound is violated. Hence an inequality constraint $g(z) \leq 0$ with optimization variable $z$ is implemented by a feature $\max(0, g(z))$. The interaction feature $d(\cdot)$ is also implemented as a quadratic penalty. We evaluate the collision avoidance features at every discrete time step in the RPI experiments. In all experiments, agent geometries are approximated by circles of radius $r_i$, which is accurate for the mobile robot and pedestrian but an over-approximation for vehicles (CARLA, exiD, Waymo) and especially for trucks in the highway exiD environment, where we use $r_i = L/2$. $L$ is the length of a vehicle. Hence, we evaluate collision avoidance every fifth timestep in the exiD, and Waymo experiments. Future work could also use more accurate vehicle approximations (e.g., multiple circles [74]) to further evaluate collision avoidance at every time step to increase the predictive performance at a higher runtime and memory cost. In the RPI experiments, we set $r_i = 0.25\,\text{m}$.

**Optimization.** As the approach already predicts accurate initial strategies $\mathbf{U}^{\text{init}}$, our experiments only required a few optimization steps. Concretely, the results of Tab. 2 and 3 in the main paper and Tab. 5 in App. G.4 are obtained with $s = 2$ optimization steps, rendering our approach real-time capable (see Fig. 15). Note while the approach also works, with a higher number of optimization steps (see Fig. 13), our experiments showed that fewer optimization steps lead to similar results, with decreased runtime and memory requirements due to the predicted initialization. The experiments (RPI, exiD, Waymo) use a stepsize of $\alpha = 0.3$ and a damping factor of $dp = 10$ in the Levenberg-Marquardt solver [32]. In the CARLA experiments, we use $s = 20$ and $\alpha = 1$. Especially in the low sample regime, a higher number of optimization steps is beneficial due to the inductive bias from the game-theoretic optimization as the initialization performance is decreased, as also later shown in Tab. 6.

### F.4 Training Details

**Loss Functions.** The imitation loss in our experiments is the minSADE [26, 50] given by:

$$\mathcal{L}^{\text{imit}} = \min_{m=1}^{M} \frac{1}{N} \sum_{i=1}^{N} \|\mathbf{x}_i^m - \mathbf{x}_{\text{GT}}\|^2 \tag{10}$$

It first calculates the average over all distances between agent trajectories $\mathbf{x}_i^m$ from agent $i$ and mode $m$ and the ground truth $\mathbf{x}_{\text{GT}}$. Then the minimum operator is applied to afterwards backpropagate the difference of the joint scene, which is closest to the ground truth. The second loss term in the exiD and CARLA environment $\mathcal{L}^{\text{goal}}$ computes the cross entropy (CE) for the goal locations averaged over all agents

$$\mathcal{L}^{\text{goal}} = \frac{1}{N} \sum_{i=1}^{N} \text{CE}\left(\mathbf{PR}_i^{\text{goal}}, \mathbf{g}_i^*\right), \tag{11}$$

whereas $\mathbf{g}_i^*$ is the goal target closest to the ground truth goal location. Lastly, $\mathcal{L}^{\text{prob}}$ computes the cross entropy for the joint futures

$$\mathcal{L}^{\text{prob}} = \text{CE}\left(\boldsymbol{\pi}, \mathbf{x}^*\right), \tag{12}$$

whereas $\mathbf{x}^*$ is the joint prediction target closest to the ground truth joint future, estimated with the minSADE. We empirically set $\lambda_1 = 1, \lambda_2 = 0.1, \lambda_3 = 0.1$ in the multi-task loss described in the main paper.

In the RPI and exiD experiments, all approaches are trained with batch size 32, using the Adam optimizer [75]. Our models in the RPI and exiD environments use a learning rate of 0.00005 across all backbones. Note that the evaluation favors the baselines, as we performed grid searches for their learning rates, whereas our approach uses the same learning rate across all backbones (exiD). In CARLA we empirically set the batch size to 16 and the learning rate to 0.0005 for our method.

In the Waymo experiment we use a batch size of 32 and a learning rate of 0.0001 across all backbones. Further, instead of the classification loss (11), we use a regression loss (minSDFE)

$$\mathcal{L}^{\text{goal}} = \min_{m=1}^{M} \frac{1}{N} \sum_{i=1}^{N} \|\mathbf{g}_i^m - \mathbf{x}_{\text{GT,goal}}\|^2 \tag{13}$$

for the goal point prediction, which calculates the distance of the closest joint goal to the ground truth goal point $x_{\text{GT,goal}}$. We set $\lambda_1 = \lambda_2 = \lambda_3 = 1$.

## G  Additional Experiments

### G.1  Qualitative Results

This section provides extended qualitative results.

**RPI.** Fig. 7 visualizes an exemplary qualitative result of the RPI experiments. Both modes collapsed when using the V-LSTM+SC baseline (explicit strategy). In contrast, this work's implicit approach better models the multi-modality present in the demonstration. Since the dataset contains solutions of games solved with different collision-weight configurations, it can be seen that our proposed method accurately differentiates between different weightings of collisions. This finding aligns with these of [14], which discovered that implicit models could better represent the multi-modality of demonstrations.

**exiD.** Fig. 8 visualizes multi-modal predictions in a highly interactive scenario, where one car (green) and one truck (yellow) merge onto the highway. The green car performs a double-lane change. Note how our model in mode three accurately predicts the future scene evolution and also outputs reasonable alternative futures. For example, in mode one, the green car performs a single lane change, whereas the blue and red cars are also predicted to change lanes. Another multi-modal prediction is visualized in Fig. 9. Observe again how the ground truth is accurately predicted in this interactive

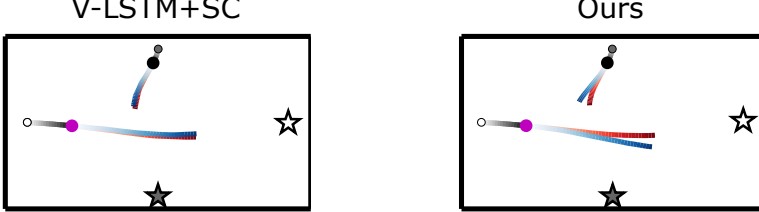

Figure 7: Qualitative comparison of the multi-modal ($M = 2$) joint predictions in the RPI environment. The start and end points of the pink agent are located on a circle with a radius of $3\,\mathrm{m}$. The start and endpoint of the black agent are visualized with a grey circle and star. The different modes are visualized by lines in red and blue color. For the baseline (V-LSTM + SC) we observe that both modes collapsed.

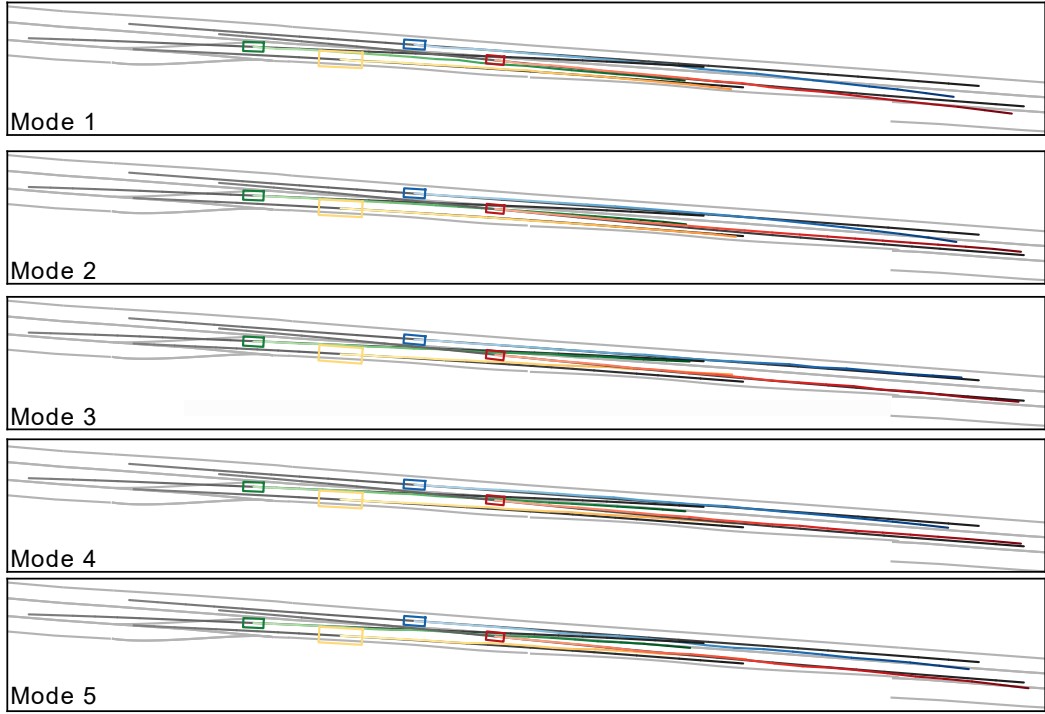

Figure 8: Multi-modal predictions in an interactive scenario (exiD, test set), in which the green and yellow perform on-ramp merges. The agent trajectories are visualized in different colors, whereas the color saturation increases the number of predicted steps. The ground truth (history and future) is shown with colors from dark grey to black and the map in light grey lines. The cars are illustrated as a rectangle. Observe how different future joint evolutions are predicted. For instance, in mode one, the red and blue vehicles change lanes to the right, whereas the green car stays in its current lane. In contrast, in mode three, the green vehicle is predicted to perform a lane change to the left, whereas the blue and red cars stay on their lane, which accurately models the ground truth.

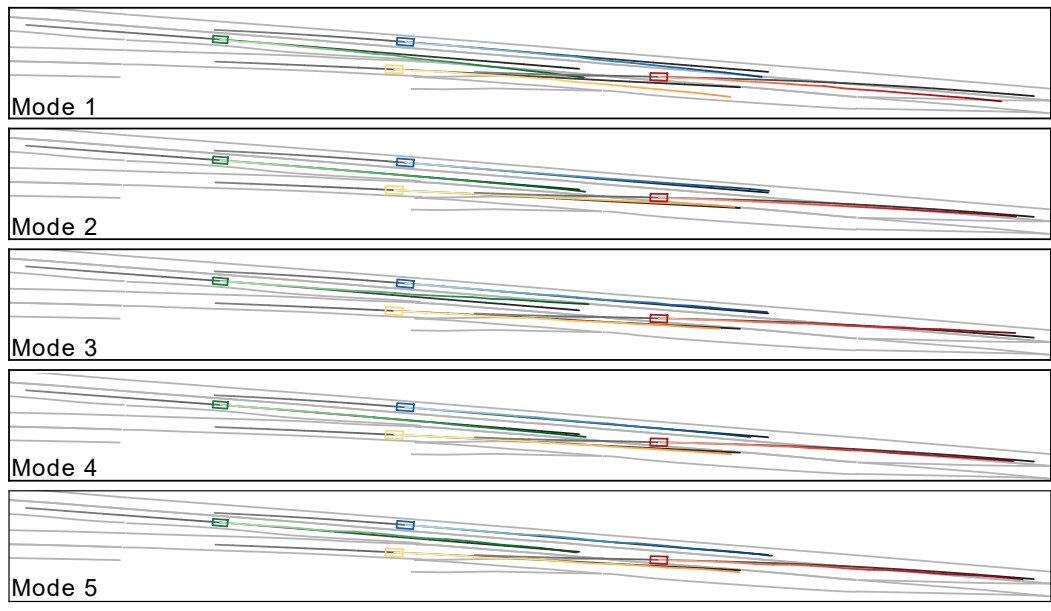

Figure 9: Multi-modal predictions in an interactive scenario (exiD, test set), in which the yellow agent merges onto the highway, and the red agent performs a double lane change. The visualization follows the description of Fig. 8. Observe how different future joint evolutions are predicted. For instance, in mode two, the ground truth is accurately predicted. In contrast, mode three represents another reasonable alternative future: The green vehicle initiates a lane change to the left. As a result, the yellow vehicle can merge with lower velocity, observable in the length of the yellow trajectory.

scenario (mode five), whereas, for example, also other plausible futures are generated. For instance, the yellow vehicle stays longer on the acceleration lane in mode one, whereas the green vehicle performs a lane change in mode three.

Fig. 10 and Fig. 11 provide visualizations of joint predictions and the interpretable intermediate representation, specifically feature weights, in more complex scenarios than those encountered in the two-player CARLA environment. In Fig. 10, it is evident that the green vehicle, in mode two, initiates braking due to the lane change executed by the blue vehicle. This lane change event is also reflected in the feature weight ($w_{\text{goal}}$) of the blue vehicle's predicted goal within the alternative lane, where a higher weight is assigned in mode two. Additionally, the resulting braking maneuver manifests as an increased weight ($w_{\text{v}}$) of the green vehicle assigned to velocity reduction in mode two. Similarly, within mode two of Fig. 11, the red vehicle embarks on a lane change towards the path of the green vehicle. Consequently, the green car begins to decelerate, which is again indicated by the weight $w_{\text{v}}$ of the green vehicle, alongside the observable alteration in the length of the green trajectory.

**CARLA.** Fig 12 visualizes another qualitative joint prediction result in the CARLA environment. Observe how our method again predicts two reasonable joint futures, whereas the correct one (mode one) has a higher probability. In mode one, the SDV (blue) goes first. That behavior is also observed when inspecting the feature weights. For example, the weight $w_{\text{vref}}$ of the SDV (blue) is higher in mode one than in mode two, inducing an acceleration in mode one. The same holds for the red agent in mode 2. Remember from the main paper the weights are normalized w.r.t. the maximum weight of the individual feature in the sequence. *Hence, weights of the same feature are comparable between the two modes, but weights from different features are not comparable.*

### G.2 Discussion of Interpretability

Following the definition of [12], interpretability in the context of SDV is achieved, among other things, by the input, output, and intermediate representations. While the input (historical trajectories and

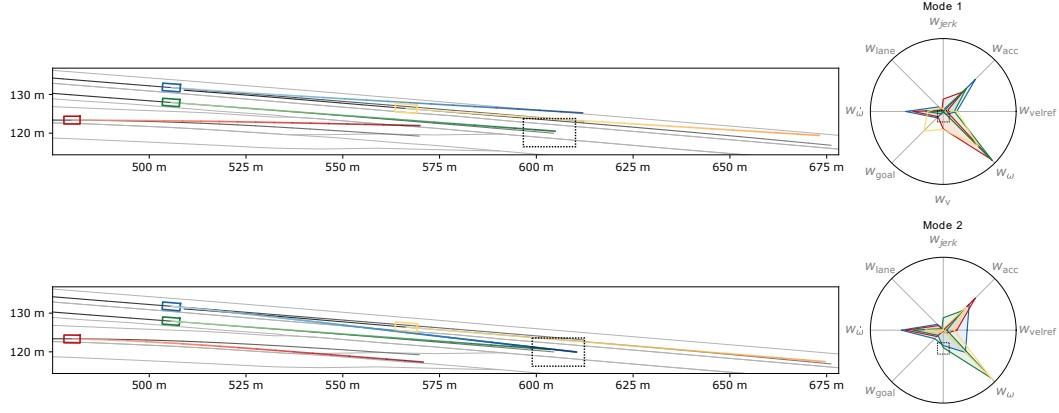

Figure 10: The two most likely joint predictions in a highway scenario (exiD, test set). The visualization follows the description of Fig. 8. Note that the weights are normalized w.r.t. the maximum weight of the individual feature in the sample. *Hence, weights of the same feature are comparable between the two modes, but weights from different features are not comparable.* The dotted black rectangles highlight important locations of the figure: Take note of the behavior in mode two, where the blue vehicle merges ahead of the green vehicle. This results in a minor braking maneuver by the green vehicle, a behavior mirrored in the (green) feature weight $w_v$, which penalizes high velocities. Conversely, in mode one, the green vehicle exhibits a swifter velocity, evident from the extended trajectory, and simultaneously maintains a lower weight for $w_v$.

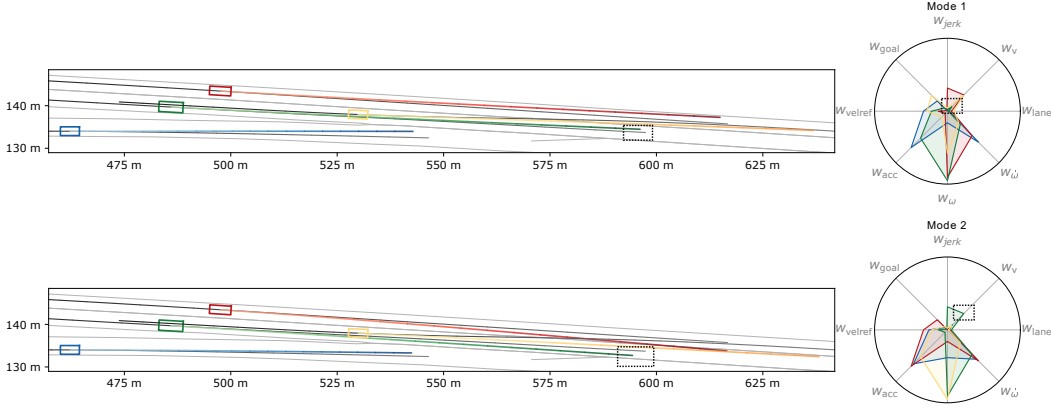

Figure 11: The two most likely joint predictions in a highway merging scenario (exiD, test set), in which the yellow agent performs a lane change. The visualization follows the description of Fig. 8. Note the behavior in mode two, where the red vehicle merges ahead of the green vehicle. Consequently, the green vehicle decelerates. That behavior is also mirrored in the elevated feature weight $w_v$, which discourages high velocities. Conversely, in mode one, the green vehicle exhibits a higher velocity, evident from the extended trajectory, and concurrently bears a low weight for $w_v$, as highlighted by the dotted rectangle in the spider plot of mode one.

map information) and the output (future trajectories) are already human-interpretable, our approach can also output interpretable intermediate representations in the form of feature weights. For instance, visualization of these features can provide a consumer in the car another instance of insights about what the SDV will do in the future. For example, assume a feature that minimizes the distance to a stopping line. A high weight could indicate that the vehicle will stop at the line. Moreover, engineers could use these weights during debugging and algorithmic design. For instance, if a weight converges to zero during training, it could indicate that the feature is unimportant. Hence, the engineer could discard the feature to reduce algorithmic complexity. Lastly, the weights could be used to design safety layers. For example, consider the scenario again in Fig. 12 and assume another module indicating if a scenario is safety-critical. If this new module now classifies that the scenario is safety-critical, while our approach plans that the SDV should accelerate (e.g., indicated by a high weight $w_{\mathrm{vref}}$), a third module could detect this conflict and overwrite the decision of our approach, to perform a braking maneuver instead.

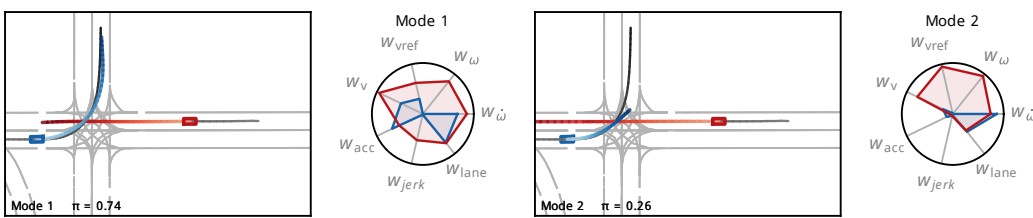

Figure 12: Qualitative joint predictions and feature weights for $M = 2$ modes (CARLA, test set). The self-dependent weights $\mathbf{W}_i^{\mathrm{own}}$ are normalized w.r.t. the maximum weight in the sequence. The ground truth is visualized with colors from dark grey to black and the map in light grey lines. Vehicles are are shown with colored rectangles and their trajectories with colored lines. $\pi$ denotes the mode probability. Observe how the blue vehicle is predicted to cross the intersection first with higher probability, which matches the ground truth. That behavior is also represented in the feature weights ($w_{\mathrm{vref}}$ is higher and $w_{\mathrm{v}}$ is lower in mode one)
.

### G.3 Ablation Studies

**Ablating the Number of Optimization Steps** The experiments revealed that the number of steps $S$ during optimization is another influential hyperparameter. Fig. 13 visualizes the impact on the minSADE. Observe how the approach gets reasonable small metrics with all configurations and hence could be used with different numbers of steps. However, while the distance between the nearest optimized joint future and the GT gets smaller with increasing optimization steps, the initialization gets slightly pushed away from the GT. Hence, with more steps, the approach gets less dependent on the initialization. [58] observes a similar effect for their differentiable single-agent optimization approach.

The previously observed behavior can have a negative influence when using a complex energy landscape with multiple local minima (e.g., when many interacting agents are present) combined with a high number of optimization steps. As the initialization gets pushed away from the ground truth, the optimizer can get stuck in local minima. Future work, could address this with different cooperative training techniques (e.g., [9]) or auxiliary losses that minimize the distance between the initialization and the ground truth.

### G.4 Quantitative Results

**Statistical Analysis.** We performed an additional statistical analysis, in which we trained our model and the baseline (`Backbone+SC`) model with three different random seeds for two different backbones (V-LSTM and VIBES). The results of all runs are visualized in Fig. 14 and the *mean and standard deviation* of the metrics over all three runs are illustrated in Tab. 5. Using the V-LSTM backbone, our model improves the baseline without overlapping error bars on the exiD dataset. However,

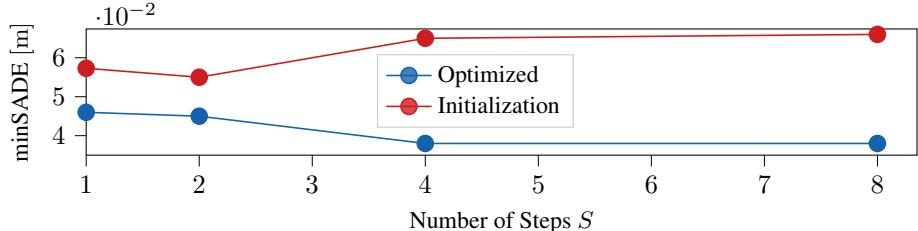

Figure 13: Predictive performance of the initial (red) and optimized strategy (blue) as a function of the number of optimization steps on the RPI validation dataset.

with the VIBES backbone, the results are not significant for all metrics and error bands overlap. We hypothesize that this is also the result of the used exiD dataset, and the improvement will be more significant on more diverse urban datasets, and additionally considering longer predictions horizons. While the exiD dataset contains many interactive scenarios, most of the vehicles in the samples perform a constant velocity movement. The competitive performance of the CV velocity model provides evidence for that claim. Moreover, in the urban CARLA experiments (6 s prediction horizon), our work improves the baseline with greater clarity. For instance, the minSADE improves by a factor of 3.88, as shown in Tab. 6. However, the CARLA dataset has a limited sample size. Filtering more interactive scenarios in exiD is impractical as the result would also be a too-small dataset. Hence, we conduct additional experiments using the interaction split of the Waymo dataset [20] (8 s prediction horizon, vehicles only). The results are also illustrated in Tab. 5 and Fig. 14 (third and fourth row). We can observe that the improvement in the distance-based metrics (minADE, minFDE, minSADE, minSFDE) is significant and also, in some metrics, one order of magnitude higher than in the exiD environment using both backbones. For instance, although the disparity in the metric minSADE between the baseline and our model is less than $0.1 \, \mathrm{m}$ on the exiD dataset, this discrepancy increases to approximately $0.5 \, \mathrm{m}$ in the Waymo dataset (a fivefold enhancement in the difference). Furthermore, the enhancement in the metric minSFDE (VIBES backbone) is notably more significant, with a factor of 27.75, on the Waymo dataset compared to the exiD dataset. Comparing the overlap rate, we perform similarly to the baseline in both experimental environments but do not improve it significantly. Future work could apply more accurate vehicle geometry approximation, with an evaluation of the collision avoidance constraint at every time step to improve the overlap rate. We can conclude the following: *Our approach improves the baselines in the joint distance metrics minSADE and minSFDE, whereas the significance is greater in more diverse urban scenarios.*

**CARLA.** Tab. 6 illustrates the predictive performance on the CARLA test set. We observed that the strongest baseline, which also uses the scene-consistent loss formulation with control prediction, does not produce reasonable predictions, despite performing grid searches for different hyperparameters. Our method outperforms the baseline by a large margin. Nevertheless, although our method demonstrates practicality in this small-sample regime, the significance of the results is comparatively limited, as previously stated in the work of [2]. Especially in automated driving applications, one has access to large datasets [20]. However, in other robotics applications such as human-robot manipulator collaborations [76], datasets are currently fairly limited, and we hypothesize that the approach could be beneficial here. We leave these studies for future work.

**Runtime and Learnable Parameters.** Training and evaluation were performed using an AMD Ryzen 9 5900X and a Nvidia RTX 3090 GPU. Fig. 15 shows the runtime dependency by varying the number of optimization steps $S$, the modes $M$, and the number of agents $N$. Our approach scales well with the number modes as all optimizations are parallelized on the GPU. The runtime increases with higher numbers of steps and more agents, as commonly reported in the game-theoretic literature. *Note that our multi-modal implementation scales better than common game-theoretic uni-modal solvers from the literature [3, Figure 4] for a higher number of agents.*

Tab. 7 compares the runtime and the number of learnable parameters for the baselines in the exiD environment. Observe how the runtime is faster than the TFL [2] baseline, which requires much

Table 5: Statistical analysis using the V-LSTM and VIBES backbone on the exiD (test set) and the Waymo (interactive validation set, vehicles only) datasets. ADE, FDE, SADE, and FDE are computed as the minimum over $M = 5$ (exiD) or $M = 6$ (Waymo) predictions in $[\mathrm{m}]$. We report the *mean and standard deviation* of the metric over three runs, which were initialized with different random training seeds. **Bold** marks the best result of each group of approaches, which uses the same observation encoding backbone. Lower is better.

| | Method | Marginal ↓ | | Joint ↓ | | |
| | | ADE | FDE | SADE | SFDE | OR |
|---|---|---|---|---|---|---|
| exiD | V-LSTM + SC | $0.84 \pm 0.01$ | $2.00 \pm 0.01$ | $1.08 \pm 0.02$ | $2.67 \pm 0.06$ | $0.010 \pm 0.001$ |
| | V-LSTM + Ours | $\mathbf{0.81 \pm 0.01}$ | $\mathbf{1.90 \pm 0.01}$ | $\mathbf{0.99 \pm 0.02}$ | $\mathbf{2.38 \pm 0.05}$ | $\mathbf{0.008 \pm 0.001}$ |
| | VIBES + SC | $\mathbf{0.73 \pm 0.01}$ | $\mathbf{1.71 \pm 0.02}$ | $1.01 \pm 0.01$ | $2.48 \pm 0.01$ | $\mathbf{0.007 \pm 0.001}$ |
| | VIBES + Ours | $0.81 \pm 0.01$ | $1.96 \pm 0.03$ | $\mathbf{1.00 \pm 0.01}$ | $\mathbf{2.44 \pm 0.04}$ | $0.008 \pm 0.001$ |
| | CV | 1.16 | 2.87 | 1.16 | 2.87 | 0.007 |
| Waymo | V-LSTM + SC | $3.08 \pm 0.04$ | $6.75 \pm 0.09$ | $3.62 \pm 0.04$ | $8.50 \pm 0.08$ | $0.047 \pm 0.002$ |
| | V-LSTM + Ours | $\mathbf{2.62 \pm 0.01}$ | $\mathbf{5.83 \pm 0.06}$ | $\mathbf{3.10 \pm 0.01}$ | $\mathbf{7.35 \pm 0.04}$ | $\mathbf{0.045 \pm 0.004}$ |
| | VIBES + SC | $3.04 \pm 0.01$ | $6.83 \pm 0.02$ | $3.53 \pm 0.04$ | $8.37 \pm 0.09$ | $\mathbf{0.042 \pm 0.001}$ |
| | VIBES + Ours | $\mathbf{2.58 \pm 0.01}$ | $\mathbf{5.84 \pm 0.03}$ | $\mathbf{3.03 \pm 0.01}$ | $\mathbf{7.26 \pm 0.04}$ | $0.047 \pm 0.002$ |
| | CV* | 10.70 | - | 10.70 | - | - |

\* Values are reported from [20].

more optimization steps due to the use of the IFT. The other baselines (V-LSTM, V-LSTM + SC) are faster, as they don't use a differentiable optimization layer, which accounts for most of the runtime (see Ours (Opt)). The V-LSTM + SC model has more parameters than the V-LSTM model, as the scene probability decoder takes in the predicted joint trajectories of all agents and has a higher depth. In contrast, the V-LSTM model can parallelize the computation as it predicts marginal probabilities per agent. Our approach also has slightly more parameters than the V-LSTM + SC baseline due to the additional game parameter and goal decoder. However, we also investigated a baseline (VIBES + SC) on the Waymo dataset, which has more parameters (1,183,687) than our model (852,786). We observed that *the baseline model with more parameters did not outperform our model with less parameters* in predictive performance. Hence, our significantly better results on the Waymo dataset (see Tab. 5) can be attributed to our system architecture and not the increased number of parameters.

## H  Additional Limitations

In this section, we name additional limitations. Our CARLA experiments are limited by the dataset size (see discussion in Sec. G.4) and can be regarded as proof-of-concept for closed-loop control. Moreover, we observed that in many scenarios, besides the interacting vehicles, other cars in the exiD dataset performed a nearly constant velocity movement, verified by the good results of the constant velocity baselines in Tab. 2 of the main paper. Further, our approach assumes an object-based environment representation with handcrafted input features (e.g., 2-D position information in agent histories) and low measurement uncertainties. However, raw-sensor data includes important information (e.g., the head movement of a pedestrian), which is relevant for downstream tasks such as motion forecasting and control. As our approach is fully differentiable, future work should explore joint perception and game-theoretic planning approaches. Doing so would allow propagating uncertainties through the whole system architecture, which has proven effective in prior work such as [36].

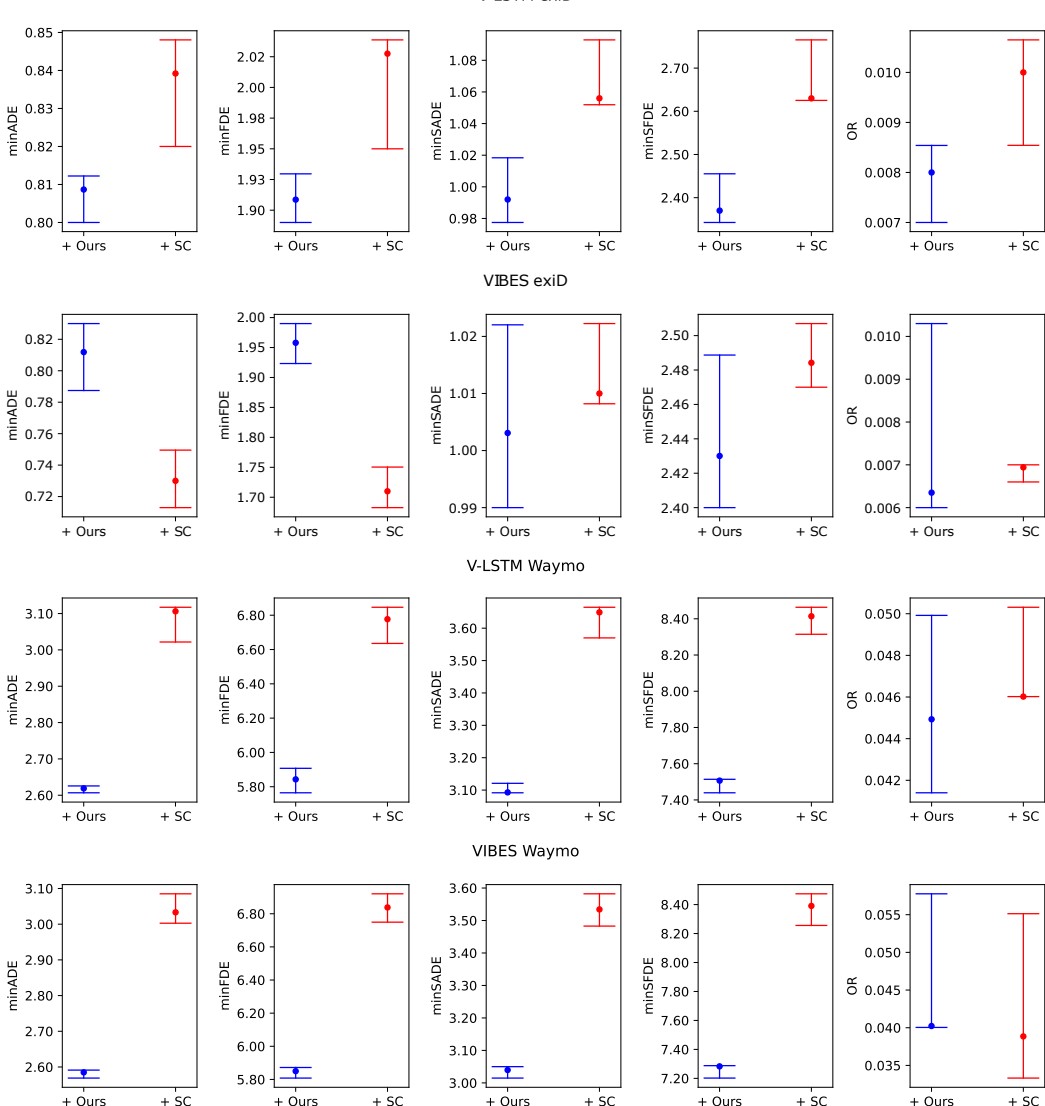

Figure 14: Error bands for the different models (red: `Backbone + SC`; blue: `Backbone + Ours`) by training each model with three different random seeds. The three results on the exiD test set and Waymo interactive validation set are marked with horizontal lines (minimum and maximum values) and a point (median). Our work on the exiD dataset improves the baseline in the joint metrics minSADE and minSFDE using the V-LSTM backbone. Using the VIBES backbone, our model performs better on average in these joint metrics, but the error bands overlap. Hence, results are not significant. Observe how our model outperforms the baseline significantly on the more challenging and diverse urban dataset (Waymo) in all distance metrics (minADE, minFDE, minSADE, minSFDE) without overlapping error bands. In both environments, our model performs on par in the OR metric.

Table 6: Predictive performance of different methods on the CARLA test dataset. The metrics and formatting are the same as in Tab. 2 in the main paper, but $M = 2$.

| Method | Marginal ↓ | | Joint ↓ | |
|---|---|---|---|---|
| | ADE | FDE | SADE | SFDE |
| V-LSTM + SC | 6.11 | 12.37 | 6.21 | 13.34 |
| V-LSTM + Ours | **1.74** | **3.28** | **1.83** | **3.43** |

Table 7: Runtime in [s] averaged over 100 exiD samples (V-LSTM backbone) and the number of learnable parameters for the baselines and our approach. Ours (NN) refers to time of the observation encoding, game parameter, and initial strategy decoding. Ours (Opt.) refers to the time for energy optimization. Ours (full) is the sum of Ours (Opt.) and Ours (NN).

|  | V-LSTM | + SC | + TGL | Ours (NN) | Ours (Opt.) | Ours (Full) |
|---|---|---|---|---|---|---|
| Runtime | 0.002 | 0.002 | 0.435 | 0.005 | 0.111 | 0.116 |
| Learnable parameters | 432,125 | 538,749 | 616,496 | 617,471 | - | 617,471 |

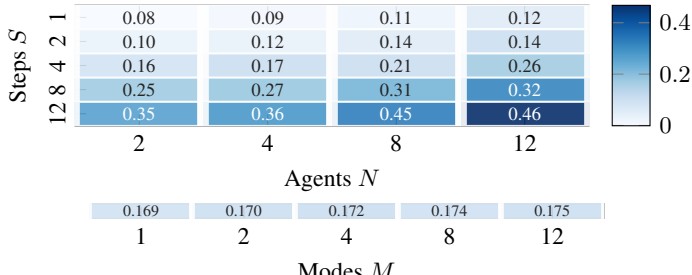

Figure 15: Runtime in [s] for different numbers of agents and optimization steps averaged over 100 exiD samples with $M = 4$. The experiments for the number of modes use $S = 4$ and $N = 4$.

