# OpenReview forum: "Energy-based Potential Games for Joint Motion Forecasting and Control"
_robot-learning.org/CoRL/2023/Conference — CoRL 2023 Poster_

### Official Review · Reviewer_xj4d · 2023-06-25

**Confidence:** 3
**Originality:** Good
**Technical Quality:** Good
**Clarity Of Presentation:** Poor
**Impact:** 2

**Recommendation:**

Weak Accept: I recommend accepting the paper, but will not argue for my recommendation if the majority of other reviewers have a different opinion.

**Review:**

Update 8/17/23: Increased score to "weak accept", as previously noted in the discussion with authors.

--------------

**Quality**: Empirically, it seems like the results are strong. It was great to see the visualizations of the predictions in the supplemental videos. I also appreciated that Table 2 shows both the positive and negative results: the proposed method does worse on the marginal metrics, but better on the joint metrics.

**Originality**:  The proposed EPOL layer is novel to the best of my knowledge. The paper derives this layer by combining ideas from prior work (differentiable optimization, potential difference games).

**Clarity**: I font the paper difficult to read, to the extent that I likely missed some of the nuances illustrated in Fig. 1 and Fig 2. A few suggestions
* In the abstract, it was unclear what problem was being solved. I think part of the trouble is that the paper is simultaneously focusing on a broad idea (multi-agent coordination) and a very specific application area (autonomous driving). One option would be to make the paper entirely about autonomous driving, and mention at the end that similar theoretical ideas might be applied to other domains. Another option would be to keep the focus more theoretical, and only introduce the autonomous driving examples in the experiments section.
* Revising the second paragraph of the introduction might help a lot with this. In particular, the intro sentence would explain that that are these two prior approaches. The next few sentences could explain the pros/cons of these approaches. And the last sentence could remain as-is (how to combine the strengths of these methods).
* In the contributions paragraph, I was unclear what the "types" of the contributions where. I.e., are they are new equation? A new neural network layer? A new algorithm for motion forecasting? (This is clarified later in the paper)
* By the end of the introduction, I was still unclear what problem was being solved by the proposed method. The end of the first paragraph suggests it's going to be something about interpretability, the second paragraph suggests it's going to be some combination of neural networks and game theory.
* The paper mentions multi-modality in a number of places, but I don't think it was ever formally defined. I'm assuming that this refers to a probability distribution with multiple modes (as opposed to a model that takes as input images + lidar + ...), but it's unclear what random variable this multi-modal distribution is modeling.
* The footnote on page 3 seems intriguing, but I didn't understand it.
* I'm a bit confused about the input to the function $C$. Eq. 2 suggests that it takes 2 inputs, an observation and a control/action. L127 uses $C$ with just one argument, the control/action. Then L141 uses it with 2 observation inputs.
* In the figures, I'd recommend making sure that the captions (1) explain what is in the figure (e.g., for Fig 3, explain that the colored rectangles refer to cars(?), and that the lines correspond to predictions); and (2) what the reader should learn from the figure.


------------------
**Minor comments**
* Run a grammar checker on the paper (e.g., Grammarly, copy-paste into a google doc)
* L38, "explicit strategies ... implicit strategies" -- I was confused by this.
* L47 -- L60: These pargraphs are great! They provide a nice description of the prior work, along with the key similarities/differences from the proposed method.
* Try to avoid starting sentences with numbered citations (L70) or equations (L121)
* Table 1: I'd recommend moving the "footnotes" to the bottom of the table, making each one a new line, and using a smaller font (perhaps \small or \tiny)
* Define notation before using it for the first time (e.g., L128 should appear before L125).
* L137 -- L144: For this section especially, it'd be good to add more intuition.
* "scene-consistent" -- Where is this defined?
* L171 -- I think ${h, m}$ should be a sequence, not a set (order does matter).
* L240 -- Add commas to the numbers ("60,338")
* L253 "unpublished model VIBES" -- Where can readers go to learn more about VIBES?
* Table 2 -- What does "+SC" refer to?
* L308 "Without loss of generality" -- Why does assuming a fixed number of agents not restrict the generality of the method?

**Quality Of The Limitations Section:**

Limitations are addressed clearly

**Questions For Rebuttal:**

1. Does the proposed method have more learned parameters than the baseline?
2. How would the proposed method compare to a baseline that uses an generic differentiable optimization layer (e.g., OptNet) instead of EPOL?

**Robotics Focus:**

Relevant but unlikely to deploy to hardware in near future

**Summary Of Paper:**

This paper proposes method for multi-agent motion forecasting based on a new neural network layer. This new layer solves a certain optimization problem (I think it's finding a certain Nash Equilibrium) while being differentiable. When this layer is plugged into prior motion forecasting systems, the resulting method achieves better _joint_ performance on the three measured benchmarks.

**Summary Of Recommendation:**

Overall, I think the ideas presented are intriguing, especially since they are backed by seem like strong empirical results. The lack of clarity in the paper makes it hard to fully appreciate the paper's contributions, especially on the theoretical front. So, while I'm currently voting to reject the paper, I would be convinced to revise my score if the paper were revised to improve the clarity.

---

> ### Author Response · Authors · 2023-08-06
> **Request for Clarification on Suggestions from Reviewer xj4d**
>
> First of all, we would like to thank you for your time and effort in writing this constructive review, which will help improve our paper's quality. We want to incorporate all your remarks and suggestion during the rebuttal. However, as the other reviewers appreciated the clarity of our paper and both state that our paper is "well-written", we would like to clarify a few points before we restructure whole sections in the manuscript.
>
> Our questions refer to the following statement in the review:
> "*I think part of the trouble is that the paper is simultaneously focusing on a broad idea (multi-agent coordination) and a very specific application area (autonomous driving). One option would be to make the paper entirely about autonomous driving and mention at the end that similar theoretical ideas might be applied to other domains. Another option would be to keep the focus more theoretical and only introduce the autonomous driving examples in the experiments section.*"
>
> Indeed our work is a mix of a theoretical contribution showing the connections of differential games, optimal control, and energy-based models and practical implementation. Both contributions were motivated by practical challenges in the game-theoretic robotics motion planning community, whose experiments were up to date mostly restricted to simulations [1-2] or simple small-scale lab [3-5]) experiments.
>
> Our intuition for introducing examples (e.g., what the individual cost terms could look like in robotics applications) in the more theoretical parts (e.g., Section 3) has been that this helps the readers' understanding by providing tangible illustrations that bridge the gap between abstract concepts and real-world applications. Moreover, keeping Section 3 more theoretical would help researchers from other robotic domains than autonomous driving (e.g., interactions between humans and robot manipulators, drones, or mobile robots) to apply the Energy-based Potential Game concept in their future work. This is particularly relevant as the cost function structures are similar across domains (e.g., penalizing the distance to a goal or high control inputs while avoiding collisions), even though the system dynamics primarily vary between these domains [4, 5].
>
> We would like to ask you if you can elaborate more in detail about suggestions for both options in general:
> - Option 1: Make the entire paper about autonomous driving
> - Option 2: Keep the paper theoretical and introduce the autonomous driving examples later
>
> Moreover, we have the following specific questions:
> - Could you name explicit lines in the theoretical parts where autonomous driving examples are introduced too early if we would follow option 2?
> - In addition, the following suggestions seem to be contrary to the suggested option 2 for keeping the whole of Section 3 theoretical as the intuition could be explained best with a practical (running) example: "*L137 -- L144: For this section especially, it'd be good to add more intuition*".
> -  If we follow option 1,  would you appreciate the following? In the revised manuscript, we will keep the theoretical structure of Section 3. However, when we introduce a practical example (e.g., referring to human preferences, cost functions, dynamics), we will introduce this in a separate paragraph at the end of a subsection (indicated with the wording "running example" similar to [6, Chapter 4]) to make the difference between the theory and the autonomous driving application more clear. In addition, we will only use examples from autonomous driving. Moreover, we will change all wording in all sections from the general robotics domain to the autonomous driving domain. That is motivated by the fact that the applicability of the EPO framework is mostly backed up by empirical evidence in the autonomous driving domain.
>
> We deeply appreciate your effort in improving our paper's quality and look forward to your response.
>
> [1] S. Le Cleac’h, M. Schwager, and Z. Manchester. Algames: a fast augmented lagrangian solver for constrained dynamic games. Autonomous Robots, 46, 01 2022.
>
> [2] S. Le Cleac’h, M. Schwager, and Z. Manchester. Lucidgames: Online unscented inverse dynamic games for adaptive trajectory prediction and planning. IEEE Robotics and Automation
> Letters, 6(3):5485–5492, 2021.
>
> [3] X. Liu, L. Peters, and J. Alonso-Mora. Learning to play trajectory games against opponents336
> with unknown objectives. art. arXiv:2211.13779, 2023
>
> [4] D. Fridovich-Keil, et al., Efficient iterative linear-quadratic approximations for nonlinear multi-player general-sum differential games. In 2020 IEEE International Conference on Robotics and Automation (ICRA), pages 1475–1481, 2020.
>
> [5] T. Kavuncu, A. Yaraneri, and N. Mehr. Potential ilqr: A potential-minimizing controller for planning multi-agent interactive trajectories. In Robotics: Science and Systems XVII, 07 2021
>
> [6] L. Peters, L. et al., Contingency Games for Multi-Agent Interaction, art. arXiv:2304.05483, 2023

---

> > ### Comment · Reviewer_xj4d · 2023-08-08
> > **Reviewer response**
> >
> > Dear Authors,
> >
> > Thanks for the response! Responding to the questions below:
> >
> > > Make the entire paper about autonomous driving
> >
> > One way to do this would be to put the problem statement from L169 -- L174 at the start of Section 3, saying "this is the problem that we aim to solve ... we'll try to solve it by building on top of some prior work on energy-based potential games, which we explain next."
> >
> > > Keep the paper theoretical and introduce the autonomous driving examples later
> >
> > One suggestion to do this would be to cut the related work on autonomous driving, perhaps only mentioning it at the end of the paper when you talk about applications.
> >
> > > Could you name explicit lines in the theoretical parts where autonomous driving examples are introduced too early if we would follow option 2?
> >
> > L15, L26. I think part of the issue is also that the discussion of the autonomous driving application is pretty long 3+ pages (which is fine if AV is the focus of the paper).
> >
> > > L137 -- L144: For this section especially, it'd be good to add more intuition
> >
> > I see why the suggestion was confusing -- sorry about that. The paper is fairly mathematically rich/dense, and so having intuition or a running example does seem like it would be good. Maybe one way to do it would be to present them in parallel. Discuss the abstract problem being solved by the potential games, then immediately offer AV as a running example.
> >
> > > would you appreciate the following?
> >
> > Yes, this sounds like a great approach!

---

> ### Author Response · Authors · 2023-08-13
> **Answers to Reviewer xj4d**
>
> Thank you once again for your time and effort and the thoughtful review, which improved the quality of our revised submission. Please find attached our responses to the individual questions and remarks.
>
> > **Quality**: Empirically, it seems like the results are strong. It was great to see the visualizations of the predictions in the supplemental videos. I also appreciated that Table 2 shows both the positive and negative results: the proposed method does worse on the marginal metrics, but better on the joint metrics.
> > **Originality**: The proposed EPOL layer is novel to the best of my knowledge. The paper derives this layer by combining ideas from prior work (differentiable optimization, potential difference games).
>
> We appreciate your positive feedback regarding results and novelty and are also looking forward to future work in this new research direction.
>
> >In the abstract, it was unclear what problem was being solved.
>
> Thank you for pointing that out. We agree that the abstract does not directly state the problem, which is solved. Hence, we added this information and restructured the abstract.
>
> >I think part of the trouble is that the paper is simultaneously focusing on a broad idea (multi-agent coordination) and a very specific application area (autonomous driving). One option would be to make the paper entirely about autonomous driving, and mention at the end that similar theoretical ideas might be applied to other domains. Another option would be to keep the focus more theoretical, and only introduce the autonomous driving examples in the experiments section.
>
> Following our previous discussion during the rebuttal, we have implemented your proposal for option two. Hence, we made the paper entirely about autonomous driving. As clarified in our previous discussion, we make that more explicit on the one side by revising the abstract and introduction. In the introduction, we also introduced a new subfigure Fig. 1a, which visualizes our problem in the autonomous driving domain at the beginning of the paper. Moreover, we changed our wording and focus from the general robotics domain to the autonomous driving domain in Sec. 1-3. As previously discussed, we kept Section 3 more theoretical. Still, we introduced an SDV running example such that the reader gets an intuition for the more abstractly described potential game and energy-based model formulation. We hope that our changes address all your clarity concerns. Otherwise, we would be happy to integrate further suggestions.
>
> > Revising the second paragraph of the introduction might help a lot with this. In particular, the intro sentence would explain that that are these two prior approaches. The next few sentences could explain the pros/cons of these approaches. And the last sentence could remain as-is (how to combine the strengths of these methods).
> > [...] By the end of the introduction, I was still unclear what problem was being solved by the proposed method. The end of the first paragraph suggests it's going to be something about interpretability, the second paragraph suggests it's going to be some combination of neural networks and game theory.
>
> Thank you for pointing that out. We implemented your suggestions by revising the paragraphs in the introduction and adding new information. In the revised manuscript, we first introduce the problem to solve, which we also illustrate in a graphical example (Fig. 1a). Then, we name the two approaches with their corresponding advantages and disadvantages. Lastly, we state that we combine the strengths of both methods.
>
>
> > I'm a bit confused about the input to the function. Eq. 2 suggests that it takes 2 inputs, an observation and a control/action. L127 uses- with just one argument, the control/action. Then L141 uses it with 2 observation inputs.
>
> You are right. That is an inconsistent notation. We corrected this in the revised manuscript in Definition 3.1. Thank you for pointing that out.
>
> > In the figures, I'd recommend making sure that the captions (1) explain what is in the figure (e.g., for Fig 3, explain that the colored rectangles refer to cars(?), and that the lines correspond to predictions); and (2) what the reader should learn from the figure.
>
> We added additional descriptions in the captions. As we introduced new figures in the supplementary material, we also followed your advice in their captions.
>
> > Run a grammar checker on the paper (e.g., Grammarly, copy-paste into a google doc)
>
> We ran a grammar check on the paper as suggested and corrected the parts in the document. Note that minor grammatically corrections, such as commas, are not highlighted in blue in.
>
> Continued to Answers to Reviewer xj4d (2/5)

---

> > ### Author Response · Authors · 2023-08-13
> > **Answers to Reviewer xj4d (2/5)**
> >
> >
> > > In the contributions paragraph, I was unclear what the "types" of the contributions where. I.e., are they are new equation? A new neural network layer? A new algorithm for motion forecasting? (This is clarified later in the paper)
> >
> > Indeed our paper has multiple contributions that should be better highlighted. First, we show how to derive an energy-based model formulation from a differential game formulation. The deviation from differential games to potential differential games has been shown in prior work, which we also highlighted as the deviation is located in Sec. 3.1 "Background." Our more theoretical contribution is how to derive an energy-based model from the potential differential game formulation (Sec. 3.2). Moreover, while prior work (see Tab. 1 of the revised manuscript) used EBMs [6,7] or differentiable game-theoretic potential game techniques [8], they did not seem to recognize that they are solving a potential game [6,7] or their learned cost could be interpreted as energy [8]. By formally highlighting these connections in our work, we hope researchers across domains can use transferable concepts in future work. Second, the EPOL layer, derived from the EPO formulation, is also novel. Lastly, our system architecture is also new, where we, as the first, combine SOTA neural networks with a nonlinear differentiable gradient-based game-theoretic optimization layer (EPOL) and evaluate the approach on real-world data. We clarified that in a revised contribution section.
> >
> >
> >
> >
> > > The paper mentions multi-modality in a number of places, but I don't think it was ever formally defined. I'm assuming that this refers to a probability distribution with multiple modes (as opposed to a model that takes as input images + lidar + ...), but it's unclear what random variable this multi-modal distribution is modeling.
> >
> > The point of multi-modality should be made more explicit. The goal is to estimate a conditional probability distribution $p(\mathbf{X} | \mathbf{o})$, whereas $\mathbf{o}$ describes the observed context (random variable), and $\mathbf{X}$ the random variable of all agents' future joint state trajectories.
> > To make the approach tractable, we marginalize over agents' joint intents to get $M$ modes of future joint strategies $p(\mathbf{X} | \mathbf{o}) = \sum^M_{m=1} \pi^m\left(\mathbf{o}\right) \, p(\mathbf{x}^m | \mathbf{o})$, which describes a mixture distribution with mixture weights given by $\pi^m\left(\mathbf{o}\right)$.
> > Here, similar to many prediction works [1,2], we model uncertainty over the discrete joint modes $M$ with a softmax distribution $\pi^m\left(\mathbf{o}\right)=\frac{\exp (f_m(\mathbf{o}))}{\sum_j \exp (f_j(\mathbf{o}))}$, whereas $f_m(\mathbf{o})$ is the output of a neural network.
> >
> > A visual example is provided on the right of Fig. 2 of the main paper, where a mode corresponds to a joint scene evolution in the future. In the example, one mode corresponds to the future scene evolution (joint trajectory), in which the white vehicle goes first. The other mode corresponds to the future scene evolution that the grey car will go first.
> >
> > We receive $p(\mathbf{x}^m | \mathbf{o})$ as follows: First we model a distribution $p(\mathbf{u}^m | \mathbf{o})$. Here, $\mathbf{u}^m$ is received by optimizing the energy function parametrized by parameters predicted by the neural network based on observation $\mathbf{o}$.
> > Moreover, the joint strategy $\mathbf{u}^m$ and the joint future state trajectory $\mathbf{x}^m$ are connected by a deterministic function $\mathbf{x}^m=f_\textrm{unroll}(\mathbf{x_0},\mathbf{u}^m)$, which unrolls the deterministic system dynamics from the current (measured) state $\mathbf{x_0}$ using the predicted strategy $\mathbf{u}^m$. That could be written as the following dirac distribution $p(\mathbf{x}^m|\textbf{u}^m,\mathbf{o}) = \delta \left(\mathbf{x}^m - f_\textrm{unroll}(\mathbf{x_0},\mathbf{u}^m)\right)$ and hence using the law of probability $p(\mathbf{x^m}|\mathbf{o})= \int p(\mathbf{x}^m|\textbf{u}^m,\mathbf{o}) \cdot p(\mathbf{u^m}|\mathbf{o}) \, d\mathbf{u}^m$.
> > By plugging together the relationships, we finally receive
> >
> > $p(\mathbf{X}|\mathbf{o})= \sum^M_{m=1} \pi^m\left(\mathbf{o}\right) \, p(\mathbf{x}^m|\textbf{o})$
> >
> > $p(\mathbf{X}|\mathbf{o})= \sum^M_{m=1} \pi^m\left(\mathbf{o}\right) \, \int p(\mathbf{x}^m|\textbf{u}^m,\mathbf{o}) \cdot p(\mathbf{u^m}|\mathbf{o}) \, d\mathbf{u}^m$
> >
> > $p(\mathbf{X}|\mathbf{o})= \sum^M_{m=1} \pi^m\left(\mathbf{o}\right) \int \delta(\mathbf{x}^m - f_\textrm{unroll}(\mathbf{x_0},\mathbf{u^m}) \cdot p(\mathbf{u^m}|\mathbf{o}) \, d\mathbf{u}^m$
> >
> > In the introduction, we described what we mean with the word $\textit{mode}$ and further extended our problem formulation in Section 3. A more detailed formulation, as described above, is also given in a new Section in App. D.
> >
> > Continued to Answers to Reviewer xj4d (3/5)

---

> > > ### Author Response · Authors · 2023-08-13
> > > **Answers to Reviewer xj4d (3/5)**
> > >
> > >
> > > > The footnote on page 3 seems intriguing, but I didn't understand it.
> > >
> > > In that particular footnote, we aimed to highlight the concept of "open-loop strategies" and their relationship to controls over time intervals. Open-loop strategies refer to a class of strategies in which the chosen controls are predetermined for all time instants, meaning that the strategy doesn't depend on the current state or the controls taken in previous time steps. In contrast to open-loop strategies, closed-loop strategies dynamically adapt controls based on the current state and the outcomes of previous controls. In other words, real-time feedback and the evolving environment influence the selection of controls in a closed-loop strategy. While closed-loop strategies could model interaction more accurately, deriving an equilibrium solution is non-trivial. However, in the spirit of model-predictive control, one can solve the game in open-loop repetitively with a high frequency to account for these modeling errors, as also stated in [3,4].
> > >
> > > Moreover, we use the same symbol $\mathbf{u}$ for the control and the strategy to avoid confusion, which is reasonable in the open-loop setting, as the strategy is a sequence of controls without state dependence. In the closed-loop setting, the strategy would be a function of $\mathbf{x}$, and introducing another variable for the strategy would be beneficial for understanding to further differentiate between the strategy and the controls. We apologize for any confusion caused by our explanation in the footnote. Please let us know if you would like us to address any specific aspects of the footnote in more detail.
> > >
> > >
> > > > L38, "explicit strategies ... implicit strategies" -- I was confused by this.
> > >
> > > The words "explicit" and "implicit" refer to the type of functions used by the different approaches. Assume $x$ is the input (observation) of a feed-forward neural networks $f_\theta(\cdot)$ predicting another variable $y$. Hence, we have an explicit relationship $y=f_\theta(x)$. In contrast, implicit functions use the relationship $y^*=\arg \min_{y} f(x,y)$. Here, we search for an optimal $y$, which together with $x$ minimizes the function $f(\cdot)$. That is a typical structure in many optimal control or game-theoretic formulations, where $f$ describes the cost. Moreover, when we parametrize $f(\cdot)$ with learnable parameters $\theta$ (e.g., by an neural network), $f_\theta(\cdot)$ describes an energy. Hence, we search for the answer $y$, which is most compatible (has the lowest energy) with the observation $x$, as also described in [5]. These implicit functions have advantageous properties, such as modeling multi-modality in demonstrations and discontinuities with the expense of a higher runtime [5, 6]. We added the information that we refer to "explicit" and "implicit" functions in Sec. 1.
> > >
> > > > L47 -- L60: These pargraphs are great! They provide a nice description of the prior work, along with the key similarities/differences from the proposed method.
> > >
> > > Thank you very much for the kind words.
> > >
> > > > Try to avoid starting sentences with numbered citations (L70) or equations (L121)
> > >
> > > We added transition words or sentences all over the document to avoid starting sentences with numbered citations or equations.
> > >
> > > > Table 1: I'd recommend moving the "footnotes" to the bottom of the table, making each one a new line, and using a smaller font (perhaps \small or \tiny)
> > >
> > > We implemented your suggestion.
> > >
> > > > Define notation before using it for the first time (e.g., L128 should appear before L125).
> > >
> > > Thank you again for pointing that out. We corrected that in the document.
> > >
> > > > L137 -- L144: For this section especially, it'd be good to add more intuition.
> > >
> > > We added more intuition for this in the SDV running example in Section 3.1. Moreover, we further provided references for the claims that this cost function can describe human behaviors and that the assumptions are reasonable for autonomous driving applications.
> > >
> > > > "scene-consistent" -- Where is this defined?
> > >
> > > We added a reference to the literature, whose definition of scene-consistency we follow, the first time we use this word in the manuscript.
> > >
> > > > Table 2 -- What does "+SC" refer to?
> > >
> > > We added an explanation of the abbreviation in the main text and the nomenclature of the supplementary.
> > >
> > > > L171 -- I think- should be a sequence, not a set (order does matter).
> > >
> > > Indeed it should not be a set, but a tuple, in which the order matters. We changed the notation in the revised document. Thank you for pointing that our.
> > >
> > > Continued to Answers to Reviewer xj4d (4/5)

---

> > > > ### Author Response · Authors · 2023-08-13
> > > > **Answers to Reviewer xj4d (4/5)**
> > > >
> > > > > L253 "unpublished model VIBES" -- Where can readers go to learn more about VIBES?
> > > >
> > > > We would like to hint you at the description of the self-developed VIBES model in App F.1. We also referred to the App. F.1, right after the wording "unpublished model VIBES" in the main text. The VIBES model is the same as the V-LSTM model [11] but has an enhanced global interaction module. While the V-LSTM uses one layer of attention to model interactions between all encoded observations (agents, map polylines), the VIBES model uses a two-stage attention mechanism. It first applies the same layer of self-attention as in V-LSTM model [11], one layer of attention, to model interactions between the agents and maps, whereas the map describes the keys. Afterward, it only uses the updated agent features in another self-attention layer to capture additional agent-to-agent interactions. In this second layer, the keys are the agents themselves. We added that additional description in App. F.1. of the supplementary.
> > > >
> > > > > L308 "Without loss of generality" -- Why does assuming a fixed number of agents not restrict the generality of the method?
> > > >
> > > > We mean that this assumption is currently a restriction on the implementation side due to the requirements of fixed size optimization variables of the used Theseus library [9] rather than on the side of the method itself.  Indeed, we also tested our approach with a higher number of agents on the exiD dataset, up to $N=8$, by filtering the dataset for scenarios where eight agents are present. However, the resulting dataset was very small. $N=4$ on exiD resulted in the most samples, so we decided to show these results.  The fixed number of agents currently refers to a fixed number of maximum interacting agents. Implementation-wise, one could also construct the joint optimization with more agents (e.g., $N=8$) and set the feature weights of non-interacting or non-present agents to zero. We leave this implementation for future work. We deleted the wording "Without loss of generality" to avoid confusion.
> > > >
> > > > > L240 -- Add commas to the numbers ("60,338")
> > > >
> > > > Commas have been added in the main paper and the supplementary.
> > > >
> > > > > Does the proposed method have more learned parameters than the baseline?
> > > >
> > > > Yes, the proposed method has slightly more learned parameters than the baselines (Backbone and Backbone+SC) due to the additional game-parameter decoders. We added the numbers of the parameters for all methods in Tab. 3 App. G.5 of the supplementary material.
> > > >
> > > > We also performed an additional ablation study to investigate the impact of the number of parameters for the baseline:
> > > > For instance, the Waymo results of the new Tab. 1 of the revised supplementary material were conducted with models of the following sizes:
> > > >  - Smaller baseline model (VIBES+SC): 786,458
> > > >  - Our model (VIBES + Ours): 852,786
> > > > Here, our model also has a slightly higher number of parameters.
> > > >
> > > > We conducted another experiment, increasing the parameters for the VIBES+SC baseline on the Waymo dataset. Specifically, we doubled the width of the two layers in the global interaction module and the three layers of the strategy decoder for the baseline.
> > > > The larger baseline model VIBES+SC has 1,183,687 parameters. Hence, it also has more parameters than our model. We observed that the performance of the larger baseline did not increase significantly and is still outperformed by our model (e.g., minSADE of our model is about 0,5m lower; minSFDE of our model is about 1m lower). That provides evidence that the improvement due to the EPOL can be attributed to our architecture, where the EPOL acts as an inductive bias and not to a higher number of parameters.
> > > >
> > > > Continued to Answers to Reviewer xj4d (5/5)

---

> > > > > ### Author Response · Authors · 2023-08-13
> > > > > **Answers to Reviewer xj4d (5/5)**
> > > > >
> > > > >
> > > > > > How would the proposed method compare to a baseline that uses an generic differentiable optimization layer (e.g., OptNet) instead of EPOL?
> > > > >
> > > > > First, we would like to clarify that the EPOL is implemented using Theseus [9], which is a differentiable nonlinear optimization library and hence can also be seen as a generic differentiable optimization layer. Indeed, one author of the Theseus paper is also the first author of your mentioned reference (OptNet), which was a pioneering work in combining optimization problems in the form of quadratic programs with neural networks. Using quadratic programs is restrictive for real-world applications due to the required quadratic cost functions and affine constraints. The authors of OpNet also state in their follow-up paper [10] that OptNets "[...] do not scale to the size of MPC problems[...]". Our formulation is an optimal control problem, which is repeatedly solved. Hence, we could also see our work as an MPC problem. However, we reimplemented an additional baseline, TGL [8], which uses a generic differentiable convex optimization layer with convex cost functions (see also the response to reviewer LVGe for more information) and linear system dynamics (2-D single integrator). Tab. 2 in the revised main paper shows the results. Our approach, which uses a nonlinear formulation, outperforms TGL due to the above restrictions of the convex formulation. Moreover, like OptNet, TGL uses the implicit function theorem (IFT) to differentiate through the optimization problem. We observed that TGL needs much more optimization steps ($s=20$) and a well-modeled initialization constant velocity in our case to get good results, as learning with the IFT cannot be used to learn the strategy initializations and needs to converge to an optimal solution to give the right gradients (see supplementary material of [9], Tab. 2). Using more optimization steps also results in a higher runtime, as shown in Tab. 3 in App. G.5 of our revised supplementary material.
> > > > >
> > > > > **References for Answers (Reviewer  xj4d)**
> > > > >
> > > > > [1] B. Sapp, Y. Chai, M. Bansal, and D. Anguelov. MultiPath: Multiple probabilistic anchor trajectory
> > > > > hypotheses for behavior prediction. Conference on Robot Learning, 2019
> > > > >
> > > > > [2] T. Phan-Minh, E. Corina Grigore, F. A Boulton, O. Beijbom, E. M Wolff. CoverNet: Multimodal behavior prediction using trajectory sets. IEEE/CVF Conference on Computer Vision and Pattern Recognition, 2019
> > > > >
> > > > > [3] S. Le Cleac’h, M. Schwager, and Z. Manchester. Algames: a fast augmented lagrangian solver for constrained dynamic games. Autonomous Robots, 46, 01 2022.
> > > > >
> > > > > [4] T. Kavuncu, A. Yaraneri, and N. Mehr. Potential ilqr: A potential-minimizing controller for planning multi-agent interactive trajectories. In Robotics: Science and Systems XVII, 07 2021.
> > > > >
> > > > > [5] Y. LeCun, S. Chopra, R. Hadsell, M. Ranzato, and F. Huang. A tutorial on energy-based learning. Predicting structured data, 2006.
> > > > >
> > > > > [6] W. Zeng, S. Wang, R. Liao, Y. Chen, B. Yang, and R. Urtasun. Dsdnet: Deep structured self-driving network. In Computer Vision – ECCV 2020, pages 156–172, Cham, 2020. Springer International Publishing.
> > > > >
> > > > > [7] W. Luo, C. Park, A. Cornman, B. Sapp, and D. Anguelov. Jfp: Joint future prediction with interactive multi-agent modeling for autonomous driving. In Proceedings of The 6th Conference on Robot Learning, volume 205 of Proceedings of Machine Learning Research, pages 1457–423 PMLR, 14–18 Dec 2023.
> > > > >
> > > > > [8] P. Geiger and C.-N. Straehle. Learning game-theoretic models of multiagent trajectories using implicit layers. Proceedings of the AAAI Conference on Artificial Intelligence, 35(6):4950–4958, May 2021.
> > > > >
> > > > > [9] L. Pineda, T. Fan, M. Monge, S. Venkataraman, P. Sodhi, R. T. Q. Chen, J. Ortiz, D. DeTone, A. Wang, S. Anderson, J. Dong, B. Amos, and M. Mukadam. Theseus: A library for differentiable nonlinear optimization. In Advances in Neural Information Processing Systems,
> > > > > volume 35, 2022.
> > > > >
> > > > > [10] Brandon Amos, Ivan Jimenez, Jacob Sacks, Byron Boots, J. Zico Kolter, Differentiable MPC for End-to-end Planning and Control, Advances in Neural Information Processing Systems 31, 2018
> > > > >
> > > > > [11] S. Ettinger, S. Cheng, B. Caine, C. Liu, H. Zhao, S. Pradhan, Y. Chai, B. Sapp, C. R. Qi, Y. Zhou, Z. Yang, A. Chouard, P. Sun, J. Ngiam, V. Vasudevan, A. McCauley, J. Shlens, and D. Anguelov.
> > > > > Large scale interactive motion forecasting for autonomous driving: The waymo open motion dataset. In Proceedings of the IEEE/CVF International Conference on Computer Vision (ICCV), pages 9710–9719, October 2021.

---

> ### Comment · Reviewer_xj4d · 2023-08-13
> **Reviewer response**
>
> Dear authors,
>
> The paper is much stronger after the revisions. Thanks for all the effort put into revising the paper and addressing the concerns. I will vote for accepting the paper [1].  I look forward to learning more about the project at CoRL!
>
> Best,
> Reviewer
>
> [1] As a reviewer, I can only update my score during the second part of the review period (Aug 16).

---

### Official Review · Reviewer_ZfDq · 2023-07-11

**Confidence:** 3
**Originality:** Good
**Technical Quality:** Very Good
**Clarity Of Presentation:** Very Good
**Impact:** 4

**Recommendation:**

Weak Accept: I recommend accepting the paper, but will not argue for my recommendation if the majority of other reviewers have a different opinion.

**Review:**

This paper clearly articulates a novel method for motion-forecasting and control and demonstrates its performance on real-world datasets. There are some weaknesses in the evaluation that would be great to see addressed.
Strengths:
* paper is well written
* results outperform the baselines on tasks of interest
* ablations which clearly demonstrate the important of the learned initialization

Weaknesses:
* alternative EPO applications discussed in Table 1 are not compared against
* lack of timing comparison against baselines - how much is learning impacting the runtime during inference?

**Quality Of The Limitations Section:**

Limitations are addressed clearly

**Questions For Rebuttal:**

Timing comparison against baselines (see main review)

Results against alternative EPO applications discussed in Table 1

**Robotics Focus:**

Relevant but unlikely to deploy to hardware in near future

**Summary Of Paper:**

The authors propose a method that connects learning and game theoretic optimization by training models to provide initializations for a downstream energy based optimizer. This method enables the system to leverage the best of both worlds: strong initializations from prior data to bootstrap optimization and interpretable/controllable optimization performance using domain knowledge.

**Summary Of Recommendation:**

Overall, I recommend accept as the paper is well-written, describes a novel idea and validates its claims with experimental results.

---

> ### Author Response · Authors · 2023-08-13
> **Answers to Reviewer ZfDq**
>
> Thank you for your time, carefully reviewing our paper and acknowledging that it "is well-written, describes a novel idea, and validates its claims with experimental results." We address all your questions for the rebuttal below:
>
> > Results against alternative EPO applications discussed in Table 1
>
> Thank you for making this suggestion. We added a comparison against the EPO approach TGL [1] in the revised document. We reimplemented the approach in our framework using the Theseus library [2], our backbone architecture, and game-parameter decoders for a fair comparison. The method requires convex constraints and a linear weighted sum of convex energy/cost features. That is restrictive for general-real world scenarios. We used single integrators for the dynamics in the $x$ and $y$ dimensions. We added the results to Tab. 2 in the main paper. Observe how this baseline is outperformed by our approach, which uses nonlinear energy features and dynamics.
> Moreover, TGL uses the implicit function theorem (IFT) to differentiate through game-theoretic optimization, which cannot be used to learn the initialization [2]. For a fair comparison, we initialize the optimization of the TGL baseline with controls equivalent to constant velocity movement, which already provides a good initialization in the exiD environment (see Tab. 2, main paper). Moreover, using the IFT, achieving the correct gradients requires convergence to an optimal solution [2] of the game-theoretic minimization, which requires more optimization steps. In our case, we used $s=20$ steps with a stepsize of $\alpha=1$ for TGL, leading to the best results of that baseline. As a downside, this higher number of required optimization steps also leads to a significant increase in runtime for the baseline, as shown in Tab. 3 App. G.5 in the supplementary material. Note that, for a fair comparison, we also searched for other optimal hyperparameters of the newly added baseline.
>
> > Timing comparison against baselines (see main review)
> > [...] How much is learning impacting the runtime during inference
>
> We added a runtime comparison in Tab. 3 App. G.5 in the supplementary material. Here, the runtime is also broken down into how much time is attributed to the neural network inference and how much time is attributed to the energy optimization. We can observe that the game-theoretic optimization dominates the runtime. It should be noted that the game-theoretic optimization would take even longer in the case we would not combine it end-to-end with the neural networks, as we would need more optimization steps. For example, the new baseline [1] can not learn an initialization and needs $s=20$ steps to achieve the results (Tab. 2 in the main paper) presented in the revised version. That also increases the runtime of the newly added baseline (TGL), as shown in Tab. 3 App. G.5 in the supplementary material.
>
> Additional runtime experiments using a different number of modes, agents, and optimization steps are also located in Fig. 12 of the App. G.5. An increasing number of agents and optimization steps leads to a higher runtime, as also highlighted in the limitations section in the main paper. In contrast, the approach has nearly constant runtime when we increase the number of modes due to the parallelized energy minimization on the GPU.
>
>
> Please inform us if this sufficiently addresses all of your comments and inquiries. Thank you again very much for your suggestions.
>
> **References for Answers (Reviewer  ZfDq)**
>
> [1] P. Geiger and C.-N. Straehle. Learning game-theoretic models of multiagent trajectories using implicit layers. Proceedings of the AAAI Conference on Artificial Intelligence, 35(6):4950–4958, May 2021.
>
> [2] L. Pineda, T. Fan, M. Monge, S. Venkataraman, P. Sodhi, R. T. Q. Chen, J. Ortiz, D. DeTone, A. Wang, S. Anderson, J. Dong, B. Amos, and M. Mukadam. Theseus: A library for differentiable nonlinear optimization. In Advances in Neural Information Processing Systems,
> volume 35, pages 3801–3818. Curran Associates, Inc., 2022.

---

> > ### Author Response · Authors · 2023-08-15
> > **Answer to Reviewer ZfDq**
> >
> > As the end of the author rebuttal period approaches, we would kindly like to ask if our answers addressed all your questions.

---

### Official Review · Reviewer_LVGe · 2023-07-12

**Confidence:** 3
**Originality:** Good
**Technical Quality:** Good
**Clarity Of Presentation:** Very Good
**Impact:** 3

**Recommendation:**

Weak Accept: I recommend accepting the paper, but will not argue for my recommendation if the majority of other reviewers have a different opinion.

**Review:**

**Strengths:**

The application of learning-based methods to enhance conventional optimization-based planners is a very promising direction. Predicting context-dependent weights for the energy function and providing these to the potential user as a form of interpretable decision-making is a nice touch particularly for the autonomous driving scenarios considered in this work. The paper is well-written and the list of abbreviations in the appendix is very helpful for ease of readability.

**Weaknesses:**
- There are many instances where performance scores of the best and second-best methods are very close. Have you tested for statistical significance of these differences (e.g. non-overlapping error bands)?
- The figures showing exiD trajectories are difficult to interpret as the content appears cramped. The video is a nice addition, while it can be difficult to understand which parts to focus on for the exiD rollouts – highlighting particular aspects e.g. via red circles would be helpful.
- The related works currently do not cover model-based RL approaches that can learn to predict multi-agent interactions for applications in driving (e.g. [1, 2]) and it would be interesting to briefly discuss relations to works from this area.

[1] Schwarting, Wilko, et al. "Deep latent competition: Learning to race using visual control policies in latent space.” CoRL, 2021.

[2] Sessa, Pier Giuseppe, et al. "Efficient Model-based Multi-agent Reinforcement Learning via Optimistic Equilibrium Computation." ICML, 2022.


**Quality Of The Limitations Section:**

Limitations are addressed clearly

**Questions For Rebuttal:**

- What is the statistical significance of the difference between the reported scores of the proposed method and the baselines?
- How reasonable is it to assume symmetry of the coupling cost terms between agents? Particularly for the exiD dataset, wouldn’t one expect to observe differences between a small car trying to merge from an on-ramp into the lane currently occupied by a large truck?
- The ablation on the optimization step count is interesting, though it’s a bit surprising that 2 optimization steps are sufficient – have you considered variations of the step size alpha?
- Could you provide more examples of interpretable weight-decodings similar to Figures 3 & 7 on more complex interaction scenarios?


**Robotics Focus:**

Highly relevant to robotics but no hardware experiments

**Summary Of Paper:**

This paper proposes to warm-start trajectory optimization with learned initial strategies and learned cost weights to enable efficient multi-modal predictions in multi-agent navigation scenarios. The approach is deployed on several benchmark datasets and evaluated across several motion forecasting backbones as well as performance metrics to yield improvements over selected baselines.

**Summary Of Recommendation:**

The paper takes a promising approach of combining context-dependent learned initializations and cost weights with down-stream trajectory optimization to improve motion forecasting, and attempts to enhance interpretability of the decision making. While the statistical analysis of the results should be improved, I’m leaning towards accept.

---

> ### Author Response · Authors · 2023-08-13
> **Answers to Reviewer LVGe (1/4)**
>
> Thank you for carefully reviewing our contribution and the constructive feedback, which improved the revised version of our paper. We appreciate your positive feedback regarding the idea of our paper and address all your comments below.
>
> >There are many instances where performance scores of the best and second-best methods are very close. Have you tested for statistical significance of these differences (e.g. non-overlapping error bands)? [...] What is the statistical significance of the difference between the reported scores of the proposed method and the baselines?
>
> We agree that this should be investigated, as the results between the strongest baseline (Backbone+SC) and our methods are indeed very close on the exiD dataset.
>
> We performed an additional statistical analysis, in which we trained our model and the best-performing baseline (Backbone + SC) model with three different random seeds for two different backbones (V-LSTM and VIBES), which is a common procedure in prior work [8,9]. The results of all runs are visualized in Fig. 11 (supplementary material), and the mean and standard deviation of the metrics over all three runs are shown in Tab. 1 in the revised supplementary material. Using the V-LSTM backbone, our model improves the baseline without overlapping error bars on the exiD dataset. However, with the VIBES backbone, while on average, we perform better on some joint metrics (e.g., minSADE, minSFDE), the results are not significant, and error bands overlap. We hypothesize that this is also a consequence of the used dataset, and the improvement is more significant on more challenging and diverse urban datasets and using longer prediction horizons. While the exiD dataset contains many interactive scenarios, most of the vehicles in the samples perform a constant velocity (CV) movement. The CV model's competitive performance and exiD figures and videos provide evidence for that claim.
>
> Moreover, our work improved the baseline with greater clarity in the (simulated) urban CARLA (6 s prediction horizon) experiments. For instance, the minSADE improves by a factor of 3.88, as shown in Tab. 2 of App. G4. However, we acknowledge that the CARLA dataset has a limited sample size.
>
> The Conference on Robot Learning (https://www.corl2023.org/call-for-papers) encourages authors to provide evidence that results translate to real-world robots. Hence, we conducted additional experiments on the Waymo Interactive dataset (8s prediction horizon, vehicle interactions) to verify our hypothesis above with an additional real-world dataset collected in six US cities. The results are also illustrated in Tab. 1 and Fig. 11 of App. G.5 (third and fourth row) in the supplementary material. We can observe that the improvement in the distance-based metrics (minADE, minFDE, minSADE, minSFDE) is significant and sometimes an order of magnitude higher than in the exiD experiments using both backbones. For instance, although the disparity in the metric minSADE between the baseline and our model is less than 0.1m in the exiD dataset, this discrepancy increases to approximately 0.5m in the Waymo dataset (a fivefold enhancement in the difference). Furthermore, the enhancement in the metric minSFDE (VIBES backbone) is notably more significant, with a factor of 27.75, on the Waymo dataset compared to the exiD dataset.
> Further, observe how the CV model is significantly worse on the Waymo dataset and is outperformed by all learning-based approaches. Comparing the overlap rate, we perform on par with the baseline in both experimental environments but do not improve significantly. Future work could apply more accurate vehicle geometry approximation (e.g., three circles), with an evaluation of the collision avoidance constraint at every time step during the optimization, to improve those metrics. For instance, in the RPI example, the one-circle approximation is accurate, leading to an overlap rate of 0.00 (Tab. 3, main paper). We can conclude that our approach improves the baselines in the joint distance metrics minSADE and minSFDE. Our approach also outperforms the baseline significantly in the metrics minADE and minFDE in the more diverse urban scenarios.
>
> We added the additional explanations of our results in Appendix G.4 and revised our conclusionsin the main paper concerning this statistical analysis.
>
> Continued to Answers to Reviewer LVGe (2/4)

---

> > ### Author Response · Authors · 2023-08-13
> > **Answers to Reviewer LVGe (2/4)**
> >
> >
> > > The figures showing exiD trajectories are difficult to interpret as the content appears cramped. The video is a nice addition, while it can be difficult to understand which parts to focus on for the exiD rollouts – highlighting particular aspects e.g. via red circles would be helpful.
> >
> > We updated all the figures showing the exiD trajectories and added additional explanations in the captions to improve clarity. Thank you for the kind words about our video. We will upload the updated video with the suggested highlighting of particular aspects before the end of the rebuttal.
> >
> > > The related works currently do not cover model-based RL approaches that can learn to predict multi-agent interactions for applications in driving (e.g. [1, 2]) and it would be interesting to briefly discuss relations to works from this area.
> >
> > Indeed our work is related to model-based reinforcement learning approaches, whereas we did not point that out sufficiently in our initial paper version. Thank you for referencing these works. We added a separate subsection in App. B with a discussion about relations and differences between the references and our work. As both approaches are online approaches interacting with the environment during learning, we also added another reference [10] in this subsection, constituting an offline learning approach similar to our work.
> >
> > >How reasonable is it to assume symmetry of the coupling cost terms between agents? Particularly for the exiD dataset, wouldn’t one expect to observe differences between a small car trying to merge from an on-ramp into the lane currently occupied by a large truck?
> >
> > Thank you for this very thoughtful question. Indeed the symmetry of coupling costs between agents is a restrictive assumption, as indicated in the extended limitations section of App. H. However, it seems reasonable in many robotics applications as automated driving, as agents share common social norms. For instance, all agents try to avoid collisions in typical driving situations. Moreover, it should be noted while the absolute values for that pairwise coupling feature (e.g., collision avoidance) are assumed to be equal between two agents, the relative value of that feature to other features (e.g., inducing breaking or acceleration maneuvers) can be significantly different. For instance, one could question if we can model aggressive drivers who do not keep safety distances to other objects with the potential game assumption. However, that behavior can also be explained by other feature terms. For instance, a feature term that minimizes the disparity between the intended velocity and a (possibly elevated) reference velocity, given significant weight compared to the collision avoidance feature, may lead to the emergence of such assertive behaviors.
> >
> > Let us consider your example: A more assertive behavior can be modeled by utilizing the aforementioned highly-weighted feature for the small vehicle. Simultaneously, the goal-decoder predicts a goal, thereby identifying the reference lane as the lane of the larger truck. This combination of factors leads to an induced acceleration during the lane change. Hence, the small vehicle merges before before the large truck. In contrast, a more passive behavior of the mentioned small car, which lets the large truck pass first before merging with it, can be achieved by predicting that the reference lane is the ramp and also predicting a high weight for minimizing the velocity, inducing deceleration. Such behavior can also be observed in our newly added Fig. 7 and Fig. 8 of the App. G.5 .
> >
> > The hypothesis that the potential game can model real-world interactions is also backed up by empirical evidence from [2-4] and our work. However, we acknowledge that this formulation does not necessarily model all adversarial behaviors of agents. Our paper constitutes an initial work in a new research area combining game-theoretic optimization with neural networks, and we are excited for future work in this domain, which could relax this assumption.
> >
> > Continued to Answers to Reviewer LVGe (3/4)

---

> > > ### Author Response · Authors · 2023-08-13
> > > **Answers to Reviewer LVGe (3/4)**
> > >
> > >
> > > >The ablation on the optimization step count is interesting, though it’s a bit surprising that 2 optimization steps are sufficient – have you considered variations of the step size alpha?
> > >
> > > We were also surprised that two optimization steps were sufficient in our experiments. However, that also has the advantage of a reduced runtime and used memory, as visualized in Figure 12 in Appendix G.4 and mentioned in the used differentiable optimization framework [5]. Our intuition is that the learned initialization already provides a reasonably good solution, which is slightly refined by the two optimization steps afterward. Moreover, the integrated game-theoretic differentiable optimization also acts as an inductive bias. For instance, consider the feature $c_\textrm{goal}$, which penalizes the distance of the last position of the optimized trajectory to the predicted goal point. Now assume that the difference is high. The result is a higher gradient, which is also backpropagated in the initial strategy decoder and hence, may provide guidance for the initialization.
> > >
> > > It should be noted that our approach also works with a higher number of steps. For instance, the CARLA experiments used $s=20$ steps, and we observed that especially in CARLA, a higher number of steps leads to better results than using a low number of steps. In contrast, in the exiD experiments, a too-high number led to unstable results. Our intuition here is as follows: First, the CARLA dataset is located in a small sample regime. Specifically, the dataset was constructed using $\textit{only ten episodes}$ of interactions at the intersection. We were even surprised that our approach outperformed the baseline methods by such a large margin in this small-sample regime and learned joint strategies applicable to single-agent motion control. In contrast, the best performant baseline did not learn reasonable predictions, highlighted by the results in Tab. 2 in App. G.5. The reason is that the optimization severely restricts the solution space, which provides evidence that the integrated optimization acts as an inductive bias, instrumental in this low-sample regime.
> > > Second, in contrast to the CARLA environment ($N=2$) the exiD environment contains more agents ($N=4$). The result is a more complex energy landscape for optimization. Combining that with the phenomenon that a higher number of steps can "push the initialization away from the ground truth" (see Fig. 10 in App. G.4) can lead to problems during energy minimization, as the gradient-based optimizer can get stuck in local minima of the more complex energy-landscape.
> > >
> > > There are multiple ways to circumvent this by using different training strategies. For instance, future work could first pre-train the initialization and then freeze the weights of the encoding backbone and the initial strategy decoder and afterward train the other decoder with activated energy minimization. Moreover, there is a whole line of research in the energy-based model literature investigating different cooperative training techniques. Exemplary works are [6] and [7]. Lastly, one could add an auxiliary loss, penalizing the distance between the closest initial joint trajectory (the result of unrolling the initial joint strategy) and the ground truth. We leave this implementation for future work. We added a description of the insights and suggestions for future work in Appendix G.4.
> > >
> > > Regarding your question about variations in the step size:
> > > We investigated variations of the step size $\alpha$. Our key insights were the following: In general, $\alpha$ is a tune-able hyperparameter, which depends on the problem formulation (e.g., the cost/ energy landscape or number of optimization variables) or other hyperparameters (e.g., the number of steps $s$). Especially, nonlinear problem formulations can have multiple local minima. Hence, when using too small $\alpha$, the optimizer can get stuck in local minima. In contrast, a large $\alpha$ can lead to overshooting. We performed grid searches for the values $\alpha=\{0.1, 0.3, 1.0\}$ in our experiments. Overall, $\alpha=0.3$ was the most robust value for all experiments, whereas $\alpha=1$ worked out slightly better in the CARLA environments. As the CARLA environment only has $N=2$ agents, and as a result, the cost/ energy landscape is more trivial than in the $N=4$ scenario, the use of more optimization steps in combination with a larger step size is also reasonable.
> > >
> > > >Could you provide more examples of interpretable weight-decodings similar to Figures 3 & 7 on more complex interaction scenarios?
> > >
> > > We introduced two new figures (Fig. 7 and Fig. 8) in more complex four agent scenarios on the exiD dataset in App. *G.1* and also added a detailed description about how the observed behavior in the scene is also represented in the interpretable weight-decodings.
> > >
> > > Continued to Answers to Reviewer LVGe (4/4)

---

> > > > ### Author Response · Authors · 2023-08-13
> > > > **Answers to Reviewer LVGe (4/4)**
> > > >
> > > > We greatly value the effort you have invested in enhancing the quality and look forward to your response.
> > > >
> > > > **References for Answer to Reviewer  LVGe:**
> > > > [1] T. Kavuncu, A. Yaraneri, and N. Mehr. Potential ilqr: A potential-minimizing controller for planning multi-agent interactive trajectories. In Robotics: Science and Systems XVII, 07 2021
> > > >
> > > > [2] P. Geiger and C.-N. Straehle. Learning game-theoretic models of multiagent trajectories using implicit layers. Proceedings of the AAAI Conference on Artificial Intelligence, 35(6):4950–4958, May 2021.
> > > >
> > > > [3] W. Zeng, S. Wang, R. Liao, Y. Chen, B. Yang, and R. Urtasun. Dsdnet: Deep structured self-driving network. In Computer Vision – ECCV 2020, pages 156–172, Cham, 2020. Springer International Publishing
> > > >
> > > > [4] W. Luo, C. Park, A. Cornman, B. Sapp, and D. Anguelov. Jfp: Joint future prediction with interactive multi-agent modeling for autonomous driving. In Proceedings of The 6th Conference on Robot Learning, volume 205 of Proceedings of Machine Learning Research, pages 1457-1467. PMLR, 14–18 Dec 2023
> > > >
> > > > [5] L. Pineda, T. Fan, M. Monge, S. Venkataraman, P. Sodhi, R. T. Q. Chen, J. Ortiz, D. DeTone, A. Wang, S. Anderson, J. Dong, B. Amos, and M. Mukadam. Theseus: A library for differentiable nonlinear optimization. In Advances in Neural Information Processing Systems, volume 35, pages 3801–3818. Curran Associates, Inc., 2022.
> > > >
> > > > [6] Y. Xu, J. Xie, T. Zhao, C. Baker, Y. Zhao, and Y. N. Wu. Energy-based continuous inverse optimal control. IEEE Transactions on Neural Networks and Learning Systems, pages 1–15, 2022.
> > > >
> > > > [7] F. K. Gustafsson, M. Danelljan, T. B. Schön, Learning Proposals for Practical Energy-Based Regression, Proceedings of The 25th International Conference on Artificial Intelligence and Statistics, PMLR 151:4685-4704, 2022.
> > > >
> > > > [8] P. Karkus, B. Ivanovic, S. Mannor, and M. Pavone. Diffstack: A differentiable and modular control stack for autonomous vehicles. In Proceedings of The 6th Conference on Robot Learning, volume 205 of Proceedings of Machine Learning Research, pages 2170–2180. PMLR, 14–18 Dec 2023.
> > > >
> > > > [9] C. Tang and R. R. Salakhutdinov. Multiple futures prediction. In Advances in Neural Information Processing Systems, volume 32. Curran Associates, Inc., 2019.
> > > >
> > > > [10] C. Diehl, T. S. Sievernich, M. Krüger, F. Hoffmann, and T. Bertram. Uncertainty-aware model-based offline reinforcement learning for automated driving. IEEE Robotics and Automation Letters, 8(2):1167–1174, 2023.

---

> > > > > ### Comment · Reviewer_LVGe · 2023-08-13
> > > > > **Response to rebuttal**
> > > > >
> > > > > Thank you very much for your extensive comments and updates to the paper. I think the discussion and revisions have strengthened the paper. I'm confident to accept.

---

> > > > > > ### Author Response · Authors · 2023-08-14
> > > > > > **Response to Reviewer LVGe**
> > > > > >
> > > > > > We sincerely appreciate your recognition of our efforts in addressing the comments. We now also uploaded the revised video, highlighting important aspects and adding further descriptions as requested.
> > > > > >
> > > > > > If you find our work to be of interest to the CoRL community, we kindly request your consideration to enhance the score in Part 2 of the review process, thereby contributing to greater visibility for the paper.

---

### Author Response · Authors · 2023-08-13
**General Response (1/2)**

First, we would like to thank the reviewers and area chair for their time and comments, which helped improve the contribution's quality. We are grateful for your positive comments
regarding ideas, results, and evaluation, as well as the constructive feedback which guided the paper's improvement.

We are excited that reviewers found our paper is "a very promising direction" (R-LVGe), "well-written" (R-LVGe and R-ZfDq), that "the ideas presented are intriguing, especially since they are backed by seem like strong empirical results" (R-xj4d), which "outperform the baselines on tasks of interest" (R- ZfDq) and that the videos were appreciated (R-LVGe, R-xj4d).

We sincerely appreciate the opportunity to incorporate the remarks and resubmit the contribution in a revised form. We have addressed all reviewer's questions and concerns with rebuttal replies and paper revisions. In this general response, we will summarize significant changes we have made to the paper. The individual answers to each reviewer will be posted in the respective threads.

First, one reviewer requested an additional statistical analysis. We conducted further experiments to emphasize the statistical significance of our results, comparing our approach on top of two observation backbones to the strongest baseline we implemented on the exiD dataset. Using the V-LSTM backbone, the improvement is significant. In contrast, while our model performs better in the joint distance-based metrics using the VIBES backbone on average, the error bands overlap. We hypothesized that this is also caused by the used dataset, which besides interactive scenarios, contains many samples where agents perform a constant velocity movement over a medium-range prediction horizon (4s), and that the benefit of our method would be more significant on more diverse urban datasets with longer prediction horizons. To provide evidence for our claims, we performed an additional experiment with statistical analysis on another interactive real-world dataset (Waymo Interaction Split [1], 8s prediction horizon) investigating different observation encoding backbones. We decided to use another real-world dataset to provide evidence, as the variation and the dataset size of the simulated RPI and CARLA environments are limited. The use of the Waymo dataset is also motivated by the fact that the Conference on Robot Learning (https://www.corl2023.org/call-for-papers) encourages authors to provide evidence that results translate to real-world robots. On Waymo, we observe that our approach generalizes to more complex urban scenarios and outperforms our strongest baseline significantly. In contrast to exiD, the improvements in some distance-based metrics are an order of magnitude higher than in the exiD environment.

Two reviewers requested a comparison against a new baseline. We added a new comparison against the requested baseline in the revised version. We reimplemented the approach of [2], which can be viewed under the EPO framework, and constitutes a differentiable game-theoretic (convex) optimization-based approach. We use our observation encoding backbone and game-parameter decoder to ensure a fair comparison. The results are located in Tab. 2 of the main paper. A runtime comparison is located in Tab. 2 of the supplementary material.

One reviewer mentioned the need for more clarity. Hence, we have followed the mentioned suggestions, restructured parts of the document, and added additional information in all sections.

Some reviewers requested new qualitative and quantitative results and additional information to describe the figures and highlight important parts. We added these in the revised version of the paper.

Continued to General Response (2/2)

---

> ### Author Response · Authors · 2023-08-13
> **General Response (2/2)**
>
> (08/15: We added information about our decision to upload code.)
>
> In particular, we have implemented the subsequent modifications to our attached manuscripts:
> - We added an evaluation using the Waymo Interaction Split [1], such that our claims are now backed up by empirical evidence from four evaluation environments (two simulations and two real-world datasets).
> - We performed statistical analysis for the strongest baseline and our model using two backbone architectures on two real-world datasets (exiD, Waymo).
> - We re-implemented and evaluated a new baseline [2] using a differentiable convex game-theoretic optimization layer (Sec. 5).
> - New qualitative results were added to show more examples of interpretable weight decodings in more complex scenarios (App. G.1).
> - Further elaboration on the impact of the number of optimization steps has been included in the App. G.4.
> - We introduced a new subfigure in Fig. 1 a) in the main paper to illustrate the problem we are addressing in this work. We also formally define the problem in the introduction and the beginning of Sec. 4. An extended probabilistic formulation has been introduced to the supplementary material (App. D).
> - We undertook a comprehensive restructuring of various sections within the document, incorporating supplementary explanations to enhance overall clarity (Abstract and Sec. 1-3). Concurrently, as suggested, we realigned the paper's emphasis towards the domain of autonomous vehicles to not overclaim our contribution and to amplify the overall clarity and coherence of the paper. Following the discussion with one reviewer during the rebuttal, we also introduced a self-driving vehicle running example to make the difference between the more abstract theory (Sec. 3) and the autonomous driving application clearer.
> - A discussion about relations to model-based reinforcement learning with new references has been added in the related work section of the supplementary material (App. B).
> - Figures have been revised to improve their clarity. Moreover, we provide additional information in their captions.
> - We addressed all other minor requests.
> - The grammar and notation have been corrected.
>
> **To access the revised documents, please refer to the attached files in the respective rebuttal sections.** We highlighted changes and newly added parts with $\textcolor{blue}{blue}$ color.
>
> **Code**: We also decided to provide the code of our differentiable game-theoretic optimization together with an observation encoding backbone around the conference date to enhance reproducibility and affiliate further research. We hope that our implementation will serve as a framework for developing hybrid algorithms in this new research direction.
>
>
> We believe that the collective merits of our study have been bolstered by several enhancements, including empirical statistical analysis, experimentation with a fourth dataset, introduction of a new baseline, incorporation of supplementary information, and restructuring for better clarity. Thank you again for the thorough and constructive reviews. Please let us know if this addresses all of your remarks and questions.
>
>
> **References for General Response:**
>
> [1] S. Ettinger, S. Cheng, B. Caine, C. Liu, H. Zhao, S. Pradhan, Y. Chai, B. Sapp, C. R. Qi, Y. Zhou, Z. Yang, A. Chouard, P. Sun, J. Ngiam, V. Vasudevan, A. McCauley, J. Shlens, and D. Anguelov.
> Large scale interactive motion forecasting for autonomous driving: The waymo open motion dataset. In Proceedings of the IEEE/CVF International Conference on Computer Vision (ICCV), pages 9710–9719, October 2021.
>
> [2] Geiger and C.-N. Straehle. Learning game-theoretic models of multiagent trajectories using implicit layers. Proceedings of the AAAI Conference on Artificial Intelligence, 35(6):4950–4958, May 2021.

---

### Decision · Program_Chairs · 2023-08-30

**Decision:**

Accept (Poster)

**Comment:**

This paper presents a learning-based method for multi-agent motion forecasting that feastures a new optimization layer that combines potential theoretic initialization and energy-based optimization.

The reviewers have found that the idea is interesting and the method is well-represented. The concern on the extensiveness of evaluation has been improved throughout the rebuttal phase.